# Beating Price of Anarchy and Gradient Descent without Regret in Potential Games

**Iosif Sakos**
Singapore University of Technology and Design
iosif_sakos@mymail.sutd.edu.sg

**Stefanos Leonardos**
King's College London
stefanos.leonardos@kcl.ac.uk

**Stelios Andrew Stavroulakis**
UC Irvine
sstavrou@uci.edu

**William Overman**
Graduate School of Business, Stanford University
wpo@stanford.edu

**Ioannis Panageas**
UC Irvine
ipanagea@ics.uci.edu

**Georgios Piliouras**
Singapore University of Technology and Design
georgios@sutd.edu.sg

## Abstract

Arguably one of the thorniest problems in game theory is that of equilibrium selection. Specifically, in the presence of multiple equilibria do self-interested learning dynamics typically select the socially optimal ones? We study a rich class of continuous-time no-regret dynamics in potential games (PGs). Our class of dynamics, *Q-Replicator Dynamics* (QRD), include gradient descent (GD), log-barrier and replicator dynamics (RD) as special cases. We start by establishing *pointwise convergence* of all QRD to Nash equilibria in almost all PGs. In the case of GD, we show a tight average case performance within a factor of two of optimal, for a class of symmetric $2 \times 2$ potential games with unbounded Price of Anarchy (PoA). Despite this positive result, we show that GD is not always the optimal choice even in this restricted setting. Specifically, GD outperforms RD, if and only if *risk-* and *payoff-dominance* equilibria coincide. Finally, we experimentally show how these insights extend to all QRD dynamics and that unbounded gaps between average case performance and PoA analysis are common even in larger settings.

## 1 Introduction

Multi-agent learning often occurs on highly non-convex landscapes even when agents have perfectly aligned or common interests (Bard et al., 2020; Dafoe et al., 2020; 2021). Thus, even if learning dynamics equilibrate, their fixed points may include saddle points or local optima of poor performance (Dauphin et al., 2014). A large stream of recent work has made considerable progress in showing convergence of optimization driven learning dynamics to locally attracting points (Ge et al., 2015; Lee et al., 2019; Mertikopoulos et al., 2019) or equilibria in cooperative games (Cohen et al., 2017; Palaiopanos et al., 2017; Anagnostides et al., 2022; Leonardos & Piliouras, 2022).[1] However, non-trivial games routinely possess attracting points of vastly different performance, and this remains true, even if one restricts attention to refined and highly robust notions of equilibria, such as pure/strict *Nash equilibria* (NE) (Kleinberg et al., 2009; Flokas et al., 2020). Accordingly, these convergence results are not enough to reason about the *quality of the learning outcomes*.

This challenge, known as the *equilibrium selection problem*, is arguably one of the thorniest problems on the intersection of game theory and learning theory with multiple practical manifestation in ML systems (Dafoe et al., 2020). Standard game-theoretic approaches (Harsanyi, 1973; Harsanyi & Selten, 1988; van Damme, 1987) or *worst-case* metrics, such as the *Price of Anarchy* (PoA) Koutsoupias & Papadimitriou (1999); Christodoulou & Koutsoupias (2005); Roughgarden (2015), offer little

---

[1] For convergence to equilibria in competitive games see also (Daskalakis & Panageas, 2018; Bailey & Piliouras, 2019; Cai et al., 2022).

insight, often none at all, from a dynamic/learning perspective. The reason is that certain NE may be reachable only from very small sets of initial conditions. Thus, instead of seeking approximately optimal performance for (almost) all initial conditions, which is generally impractical, we need to develop and argue about *average performance measures* that couple the limit point of the learning process with the likelihood that such an outcome is reached by the given learning dynamics.

Based on the above, our focus is to study 1) under which conditions can we reliably compare the performance of standard learning dynamics, such as replicator and gradient descent, in games and 2) under which conditions can they be shown to be approximately optimal under random initialization even when the performance gap between different equilibria, i.e., the PoA, is unbounded.

**Model and Contributions.** To make progress in these directions, we study *Q-replicator dynamics* (QRD) in general potential games (PGs). QRD are one of the most general classes of continuous time, multi-agent learning dynamics that include *gradient descent* (GD), *replicator* (RD) and *log-barrier* dynamics as special cases (Giannou et al., 2021). On the other hand, PGs are the standard model of multi-agent coordination and include congestion and identical interests games as important sub-classes (Wang & Sandholm, 2002; Panait & Luke, 2005; Carroll et al., 2019; Dafoe et al., 2020).

To analyze the average- and comparative-performance of QRD in PGs, an essential first step is to establish their convergence to equilibrium points. In our first result, we answer this question affirmatively by proving *pointwise* convergence of all QRD to NE in almost all finite PGs (Theorem 3.2). Our proof combines recent results from Swenson et al. (2020)[2] with standard convergence techniques in the study of PGs, e.g., Palaiopanos et al. (2017); Mertikopoulos & Sandholm (2016; 2018).

In Section 4, we turn to our main focus of equilibrium selection, we restrict our attention for analytical tractability to a class of 2-agent, 2-action ($2 \times 2$) PGs. Despite its simple representation, this class retains all complexities of multi-agent learning that we aim to study. Specifically, as we show, there exist games in this class which have arbitrarily large PoA, i.e., for which the worst-case equilibrium can be arbitrarily worse than the socially optimal outcome, but for which the *Average Price of Anarchy* (APoA) under GD admits a (tight) upper bound of 2 (Theorem 4.8). Despite this positive news for GD, we also show that, even in this *restricted* class of games, GD is *not always optimal*. Specifically, GD reaches the payoff-dominant, i.e., socially optimal, equilibrium more often than RD, if and only if, this equilibrium is also risk-dominant (less risky) (Theorem 4.6). To the best of our knowledge, this provides the first comparison in the performance between two optimal no-regret dynamics and adds a new dimension to the long literature on the important interplay between risk- and payoff-dominance in game-theoretic dynamics (Kim, 1996; Schmidt et al., 2003; Sato et al., 2005; Kaisers & Tuyls, 2011; Kianercy & Galstyan, 2012; Leonardos & Piliouras, 2022; Pangallo et al., 2022).

Finally, in section 5, we present experiments in larger PGs for which risk- and payoff-dominance can be properly generalized (diagonal payoff matrices). Such games are, in fact, designed to be as hard as possible from the perspective of average case performance due to their exponentially large number of NE and unbounded PoA. However, as our experimental results rather surprisingly indicate, increasing the size of the game seems to have little to no effect on average-case or relative performance and the analogues of Theorem 4.6, i.e., comparative performance of RD and GD, and Theorem 4.8, i.e., bounded APoA (in fact typically around 1.2), continue to hold.

**Related Work** The literature on average-case performance is scarce. As exhibited by Panageas & Piliouras (2016); Zhang & Hofbauer (2015); Pangallo et al. (2022) one of the main reasons is that regions of attraction are complex geometric manifolds that become mathematically intractable even in low-dimensional settings. Average performance metrics are first introduced in Panageas & Piliouras (2016) who is the main precursor to our work. Panageas & Piliouras (2016) focus exclusively on replicator dynamics and prove pointwise convergence to NE only for the special cases of congestion games with linear cost functions and network coordination games. Their bounds on APoA only apply in restricted instances of common interest games (Stag Hunt). Critically, prior to our work it was not known whether a rigorous comparison between RD and GD is possible in any single game setting.

---

[2] Swenson et al. (2020) shows that all NEs in *almost all*, i.e., in all but a closed set with Lebesgue measure zero, PGs are regular in the sense of Harsanyi (1973), i.e., they are isolated and highly robust.

Regarding the technical tools, our approach makes use of two connected, yet fundamentally different, theories. The first part, i.e., *convergence*, relies on the theory of *Lyapunov analysis* and the properties of dissipative systems, i.e., systems that lose momentum over time till they converge to a steady state (Haddad & Chellaboina, 2008). This in itself is a standard staple of analyzing learning in potential games (Kleinberg et al., 2009; Cohen et al., 2017; Anagnostides et al., 2022). By contrast, the second part, i.e., *performance*, relies on the existence of *invariant functions*, a feature most often studied in conservative systems, that characterize stable and unstable areas in the state space of such systems (Palaiopanos et al., 2017; Nagarajan et al., 2020). It is important to note that this proof technique is scalable to higher dimensional settings. The key idea is to identify one or more (in more complex games) invariants of motion for QRD or other game dynamics. This is arguably a challenging task, but several recent works showcase that this is a viable proof strategy in rather general settings Balduzzi et al. (2018); Mertikopoulos et al. (2018); Piliouras & Wang (2021); Paik & Griffin (2023).

## 2  PRELIMINARIES: GAME-THEORETIC AND BEHAVIORAL MODELS

**Game-theoretic model.**    A multi-agent finite potential game $\Gamma := \{\mathcal{N}, (\mathcal{A}_k, u_k)_{k \in \mathcal{N}}, \Phi\}$ denotes the interaction between a set of $\mathcal{N} := \{1, \ldots, n\}$ agents. Each agent $k \in \mathcal{N}$ has a finite set of actions, $\mathcal{A}_k$, with size $|\mathcal{A}_k|$, and a reward function $u_k : \mathcal{A} \to \mathbb{R}$ where $\mathcal{A} := \prod_{k \in \mathcal{N}} \mathcal{A}_k$ is the set of all *pure action profiles* of $\Gamma$. Agents may use mixed actions or *choice distributions*, $x_k = (x_{k,a_k})_{a_k \in \mathcal{A}_k} \in \mathcal{X}_k$, where $x_{k,a_k}$ is the probability with which agent $k$ uses their action $a_k \in \mathcal{A}_k$ and $\mathcal{X}_k := \{x_k \in \mathbb{R}^{|\mathcal{A}_k|} \mid \sum_{a_k \in \mathcal{A}_k} x_{k,a_k} = 1, x_{k,a_k} \geq 0\}$ is the $(|\mathcal{A}_k| - 1)$-dimensional simplex. Given any mixed-action $x_k \in \mathcal{X}_k$, we write $\text{supp}(x_k) := \{a_k \in \mathcal{A}_k \mid x_{k,a_k} > 0\}$ to denote the *support* of action $x_k$, i.e., the set of all pure actions $a_k \in \mathcal{A}_k$ that are selected with positive probability at $x_k$. Using conventional notation, we also write $s = (s_k, s_{-k}) \in \mathcal{A}$ and $x = (x_k, x_{-k}) \in \mathcal{X} := \prod_{k \in \mathcal{N}} \mathcal{X}_k$ to denote the *joint pure and mixed action profiles* of $\Gamma$, where $s_{-k}$ and $x_{-k}$ are the vectors of pure and mixed actions, respectively, of all agents other than $k$. When time is relevant, we will use the index $t$ for all the above, e.g., we will write $x_k(t)$ for agent $k$'s choice distribution at time $t \geq 0$. Finally, a function $\Phi : \mathcal{A} \to \mathbb{R}$ is called a *potential function* of $\Gamma$ if it satisfies $u_k(s) - u_k(s'_k, s_{-k}) = \Phi(s) - \Phi(s'_i, s_{-i})$, for all $k \in \mathcal{N}$ and all $s, s' \in \mathcal{A}$. The agents' reward functions and the potential function extend naturally to mixed action profiles with $u_k(x) = \mathbb{E}_{s \sim x}[u_k(s)]$ and $\Phi(x) = \mathbb{E}_{s \sim x}[\Phi(s)]$.

**Regular Nash and restricted equilibria.**    A *Nash equilibrium (NE)* of $\Gamma$ is an action profile $x^* \in \mathcal{X}$ such that $u_k(x^*) \geq u_k(x_k, x^*_{-k})$, for all $k \in \mathcal{N}$ and for all $x \in \mathcal{X}$. By linearity of expectation, the above definition is equivalent to: $u_k(x^*) \geq u_k(a_k, x^*_{-k})$, for all $a_k \in \mathcal{A}_k$, and all $k \in \mathcal{N}$, where $u_k(a_k, x^*_{-k})$ is the reward of agent $k$ for playing pure action $a_k$ against mixed strategies $x^*_{-k}$ by the remaining agents. Let $\text{NE}(\Gamma)$ denote the set of all NE of $\Gamma$. A NE is called symmetric if $x_1^* = \ldots = x_n^*$, and is called *fully mixed* if $\text{supp}(x^*) = \prod_{k \in \mathcal{N}} \text{supp}(x_k^*) = \mathcal{A}$. A NE is called *regular* if it satisfies the following definition.

**Definition 2.1** (Regular Nash equilibria (Harsanyi, 1973; Swenson et al., 2020)). *A Nash equilibrium, $x^* \in NE(\Gamma)$, is called* regular *if it is (i) quasi-strict, i.e., if for each player $k \in \mathcal{N}$, $x_k^*$ assigns positive probability to all best responses of player $k$ against $x^*_{-k}$, and (ii) second-order non-degenerate, i.e., if the Hessian, $H(x^*)$ of the potential function $\Phi$, taken with respect to $\text{supp}(x^*)$ is non-singular.*

Finally, a *restriction*, $\Gamma' := \{\mathcal{N}, (\mathcal{A}'_k, u'_k)_{k \in \mathcal{N}}\}$ of $\Gamma$, is a game where $\mathcal{A}'_k \subseteq \mathcal{A}_k$ and $u'_k : \mathcal{A}' \to \mathbb{R}$ is the restriction of $u_k$ to $\mathcal{A}' := \prod_{k \in \mathcal{N}} \mathcal{A}'_k$ for all $k \in \mathcal{N}$. We write $R_\Gamma$ to denote the set of all restrictions of $\Gamma$. An action-profile $x \in \mathcal{X}$ is called a *restricted equilibrium* of $\Gamma$ if it is a Nash equilibrium of a restriction of $\Gamma$, cf. Mertikopoulos & Sandholm (2018). It is easy to see that all restrictions of a potential game $\Gamma := \{\mathcal{N}, (\mathcal{A}_k, u_k)_{k \in \mathcal{N}}, \Phi\}$ are also potential games, whose potential functions are restrictions of $\Phi$ to the respective subspace of $\mathcal{A}$.

**Behavioral-learning model.**    The evolution of agents' choice distributions in the joint action space, $\mathcal{X}$, is governed by the *q-replicator dynamics (QRD)* which are described by the parametric system of differential equations (equations of motions) $\dot{x} := V_q(x)$, where $V_q : \mathcal{X} \to \mathbb{R}^{|\mathcal{A}|}$ is given by

$$\dot{x}_{k,a_k} = x_{k,a_k}^q \left( u_k(a_k, x_{-k}) - \frac{\sum_{a_j \in \mathcal{A}_k} x_{k,a_j}^q u_k(a_j, x_{-k})}{\sum_{a_j \in \mathcal{A}_k} x_{k,a_j}^q} \right), \quad \text{for all } k \in \mathcal{N}, a_k \in \mathcal{A}_k, \text{ (QRD)}$$

for any $q \geq 0$. Special cases of the above dynamics are the projection or *gradient descent (GD)* dynamics, for $q = 0$, the (standard) *replicator (RD)* dynamics, for $q = 1$, and the *log-barrier* or inverse update dynamics, for $q = 2$ (Mertikopoulos & Sandholm, 2018; Giannou et al., 2021).

## 3 POINTWISE CONVERGENCE OF QRD TO NASH EQUILIBRIA

Our results consist of two parts. In the first part, which is the subject of this section, we show convergence of QRD to Nash equilibria in a subclass of potential games we dub *Perfectly-Regular Potential Games (PRPG)* that contains *almost all* finite potential games.

**Definition 3.1** (Perfectly-regular potential games)**.** *A potential game $\Gamma$ is called* regular *if it has only regular Nash equilibria. A regular potential game is called* perfectly-regular potential game *(PRPG) if all its restrictions are regular potential games, i.e., if they only possess regular Nash equilibria.*

As we show in Lemma B.2 in the Appendix, almost all finite potential games are PRPGs; this is a direct generalization of Swenson et al. (2020) who prove that almost all potential games are regular. The PRPG class contains many widely-studied subclasses of games, e.g., congestion games, or games with identical reward functions (Wang & Sandholm, 2002; Panait & Luke, 2005; Carroll et al., 2019; Dafoe et al., 2020). Examples of non-PRPG games are degenerate games with equal payoffs, e.g., a two-player game with actions $\mathcal{A}_1 = \mathcal{A}_2 = \{a_1, a_2\}$ and payoffs $u_k(a_i, a_j) = 1$ for all $i, j \in \mathcal{A}_k$ and $k = 1, 2$. The convergence result of QRD in this class is stated formally in Theorem 3.2.

**Theorem 3.2** (Pointwise convergence of QRD to NE in PRPGs)**.** *Given any perfectly-regular potential game (PRPG), $\Gamma$, and any interior initial condition $x(0) \in \mathrm{int}\, \mathcal{X}$, the q-replicator dynamics, defined as in equation QRD, converge pointwise to a Nash equilibrium $x^*$ of $\Gamma$ for any parameter $q \geq 0$. Furthermore, the set $\mathcal{Q}(\mathrm{int}\, \mathcal{X}) := \bigcup_{x_0 \in \mathrm{int}\, \mathcal{X}} \{x^* \in \mathcal{X} \mid \lim_{t \to \infty} x(t) = x^*,\ x(0) = x_0\}$, i.e., the set of all limit points of interior initial conditions, is finite.*

Importantly, Theorem 3.2 states that QRD converge to some NE for *almost all initial conditions* in *almost all potentials games*. A direct implication is that when reasoning about the quality of the collective learning outcome in cooperative multi-agent settings, as captured by PRPGs, we can restrict our attention to NE. This is the subject of the next section. Before proceeding with this, we provide a sketch of the proof of Theorem 3.2 (cf. Appendix B for the formal proof).

The proof of Theorem 3.2 proceeds in two steps, which utilize that (i) PRPGs have a finite number of regular equilibria, and (ii) the probability of optimal actions *near* an equilibrium point is increasing in time with respect to QRD. In the first step, we prove that for any initial condition, the sequence of joint action profiles, $x(t)_{t \geq 0}$, that is generated by QRD for any $q \geq 0$ converges to a restricted equilibrium of a PRPG, $\Gamma$. This relies on the fact that the set of cluster (limit) points of the trajectory—also called the $\omega$-limit set—is a finite, and in fact, as we show, a single element set for any PRPG. In turn, this holds because any PRPG provably contains only a finite number of restricted equilibria. In the second step, we show that any such limit point has to be a NE of $\Gamma$. To establish this, we exclude convergence to restricted equilibria that are not NE of $\Gamma$ by coupling the structure of PRPGs, which ensures that there is a finite number of (regular) restricted equilibria, with the nature of QRD which guarantees that in the vicinity of a limit point, optimal actions, i.e., best responses, need to be played with increasingly higher probability. As a result, all actions in the support of the limit choice distribution of each agent must be best responses against the actions of all other agents, which implies that all points that can be reached by QRD are NE of $\Gamma$.

## 4 PERFORMANCE OF THE COLLECTIVE LEARNING OUTCOME

**Beyond static performance metrics.** We next turn our attention to the main challenge of quantifying the quality of the collective learning outcome. To do that, we need to derive *appropriate* performance metrics. In static regimes, we can rely on a variety of meaningful metrics, e.g., the *Price of Anarchy (PoA)* (Koutsoupias & Papadimitriou, 1999; Christodoulou & Koutsoupias, 2005; Roughgarden, 2015), which is defined as the ratio between the *socially worst* NE of the game and the *socially optimal* outcome (in terms of the agents' sum of rewards). However, despite the useful insights that PoA provides in general games, it is not difficult to find PRPGs in which the PoA fails to provide any meaningful information about the game. Let us consider the following example:

**Example 4.1** (A simple example of unbounded performance loss). *Consider a parametric $2\times2$-PRPG, $\Gamma_w$, i.e., a 2-player 2-actions PRPG with identical payoff functions $u_{w,1}(s_1, s_2) = u_{w,2}(s_2, s_1) = A_w(s_1, s_2)$, where matrix $A_w \in \mathbb{R}^{2\times2}$ is given by $A_w = diag(1, w)$, with $1 \leq w$.[3] The game $\Gamma_w$ has two pure NE, one that corresponds to $x_1 = (1, 0)$ and $x_2 = (1, 0)$ with social welfare $\mathrm{SW}(x) = 1 + 1 = 2$, and one that corresponds to $x_1' = (0, 1)$ and $x_2' = (0, 1)$ with social welfare $\mathrm{SW}(x') = w + w = 2w$. Since $w$ can take any value larger than 1, the difference in performance can be arbitrary large with respect to the PoA. Specifically, $PoA(\Gamma_w) = \frac{\mathrm{SW}(x')}{\mathrm{SW}(x)} = w \to \infty$ as $w \to \infty$.*

While useful in static environments, the PoA metric does not capture the dynamic nature of multi-agent learning. In particular, it does not provide an answer to the question: *How likely is it for the agents to reach a good or bad outcome given that the multi-agent system converges?*

### 4.1 REGIONS OF ATTRACTION AND AVERAGE PERFORMANCE METRICS

To answer the above question and argue about the collective performance of the game dynamics, we need to quantify the likelihood of each outcome when the initial conditions of the system are randomly sampled. A *region of attraction* of a given outcome formalizes this notion.

**Definition 4.2** (Regions of attraction). *Let $\Gamma$ be any game and assume that its joint action profile, $x \in \mathcal{X}$, is evolving according to the equations of motion $\dot{x} = f(x)$. Then for any $x^* \in \mathcal{X}$, the set $RoA_{f,\Gamma}(x^*) := \{x_0 \in \mathcal{X} \mid \lim_{t\to\infty} x(t) = x^*, \; x(0) = x_0\}$ is called the region of attraction (RoA) of $x^*$ with respect to the dynamics $f$.*

In other words, the RoA of a point $x^* \in \mathcal{X}$ is the set of all initial conditions in $\mathcal{X}$ for which the dynamics asymptotically converge to $x^*$. Note that RoAs do not intersect. If we can determine the regions of attraction of some game dynamics, then given a certain static performance metric, e.g., the social welfare, we can define a corresponding *average-performance metric* that *weighs-in* all possible outcomes, in the sense of limit points, according to their likelihood of occurring with respect to the given dynamics. In order for this *average* to be *meaningful*, a minimum requirement, is that the dynamics converge for almost all, i.e., all but a measure zero, initial conditions. An Average Performance Metric (APM), refined from (Panageas & Piliouras, 2016), is defined as follows[4]:

**Definition 4.3** (Average-performance metric). *Let $\Gamma$ be a multi-agent game and assume that its joint action profile, $x \in \mathcal{X}$, is evolving according to the equations of motion $\dot{x} = f(x)$. Let $\mathcal{X}_0 \subseteq \mathcal{X}$ be a set of initial conditions such that the set of convergence points $\mathcal{Q}(\mathcal{X}_0)$ is finite. Then, given a performance metric $g : \mathcal{X} \to \mathbb{R}$ of $\Gamma$, the average-performance of the dynamics governed by $f$ in $\Gamma$ with respect to the performance metric $g$ and the set of initial condition $\mathcal{X}_0$, is given by*

$$APM_{g,\mathcal{X}_0}(f, \Gamma) := \sum_{x^* \in \mathcal{Q}(\mathcal{X}_0)} \mu(RoA_{f,\Gamma}(x^*)) \cdot g(x^*), \tag{APM}$$

*where $\mu$ is a probability measure on $\mathcal{X}_0$.*

In other words, APM is the expected optimality of a random initialization of the dynamics in $\mathcal{X}_0 \subseteq \mathcal{X}$ with respect to some metric $g$. For instance, if the performance metric $g$ is the social welfare, then the average-performance metric with respect to $g$ measures the expected social welfare of the system for any random initialization in $\mathcal{X}_0$. The average-performance metric that we are going to use in the remainder of this section is the *Average Price of Anarchy (APoA)* (Panageas & Piliouras, 2016). The APoA is an APM with respect to the social welfare, re-normalised such that the APoA is greater than or equal to 1, with equality only if (almost) all the initial conditions converge to the socially optimal outcome of the system. Formally, given a multi-agent game $\Gamma$, equations of motion $\dot{x} = f(x)$ that describe the evolution of the agents actions in $\Gamma$, and a set of initial conditions $\mathcal{X}_0 \subseteq \mathcal{X}$ that consists of almost all points in $\mathcal{X}$, the APoA is given by

$$APoA(f, \Gamma) := \frac{\max_{x \in \mathcal{X}} \mathrm{SW}(x)}{APM_{\mathrm{SW}, \mathcal{X}_0}(f, \Gamma)}. \tag{APoA}$$

---

[3]Here, $diag(a_1, a_2)$ denotes a matrix with $a_1, a_2$ on the diagonal and zeros otherwise. Also, we write both $x, y \in [0, 1]$ to denote the mixed choice distributions, $(x, 1-x), (y, 1-y)$, of players 1 and 2, respectively.

[4]Recall that a probability measure $\mu$ on a compact space $\mathcal{X}$ is countably-additive function from the powerset of $\mathcal{X}$ to $\mathbb{R}_+$ such that $\mu(\mathcal{X}) = 1$ and $\mu(\emptyset) = 0$.

Notice that a large APoA, like a large PoA, is a negative trait that depends on the game $\Gamma$, but, in contrast to PoA, it also depends on the game dynamics $f$. In other words, a lower APoA is an indication of better performance of $f$ in $\Gamma$ compared to some other dynamics with larger APoA.

**Remark 4.4.** *As we mentioned above, Definition 4.3 does not ensure that an APM is always a* meaningful *metric for the system. However, as long as one can prove that (i) the dynamics converge pointwise to some $x^* \in \mathcal{Q}(\mathcal{X}) \subseteq NE(\Gamma)$ for almost all initial condition $x_0 \in \mathcal{X}$, and (ii) the set of limit points, $\mathcal{Q}(\mathcal{X})$, is finite —two conditions that are satisfied by any PRPG that evolves with respect to some QRD (cf. Theorem 3.2)—the APoA has an intuitive interpretation. Specifically, in this setup, the APoA is always bounded between the PoA and the* Price of Stability (PoS) *of the game, i.e., the ratio between the socially optimal outcome and the socially optimal NE (Roughgarden, 2015).*

## 4.2 THE TAXONOMY OF QRD IN $2 \times 2$ PRPGS

Having established appropriate performance measures and that PRPGs are an appropriate class in which standard learning dynamics, like QRD, can be compared in terms of performance, in this section, we perform a complete theoretical analysis of our two motivating questions in the subclass of symmetric $2 \times 2$ PRPGs. The main takeaway of this analysis is that if the payoff-dominant NE requires relatively low risk for the players compared to the other NE of the game, i.e., if it is risk-dominant, then GD performs better than RD, and vice versa, if the payoff-dominant NE of the game fails to be the risk-dominant one, RD performs better than GD. To the best of our knowledge, this is the first rigorous analysis on the relative performance between GD and RD. Concerning our second question, we establish that if the payoff- and risk-dominant equilibria coincide, the APoA of GD is upper bounded by 2. It is important to note that the class of symmetric $2 \times 2$ PRPGs is a non-trivial regime for the comparison of the two dynamics, since PoA is provably unbounded in this setup, cf. Example 4.1. Omitted materials may be found in Appendix C.

**Representation of symmetric $2 \times 2$ PRPGs.** We begin this section by showing that any $2 \times 2$ symmetric PRPG is equivalent to a game $\Gamma_{w,\beta}$ as defined in Lemma 4.5. The only non-generic games that are excluded from this reformulation are *dominance-solvable* games whose analysis is trivial and, therefore, outside of our scope. To proceed with the formal statement of Lemma 4.5, recall that a NE, $x^*$, of a symmetric potential game, $\Gamma$, is called *payoff-dominant*, if $u_k(x^*) \geq u_k(x')$ for all $x' \in \text{NE}(\Gamma)$, and it is called *risk-dominant*, if $x^*$ is unilaterally optimal against the uniform distribution of the rest of the agents (Harsanyi & Selten, 1988).

**Lemma 4.5.** *Any $2 \times 2$ symmetric PRPG, $\Gamma$, with payoff functions $u_1(s_1, s_2) = u_2(s_2, s_1)$ can be equivalently represented by a game $\Gamma_{w,\beta}$ with payoff functions $u_{w,\beta,1}(s_1, s_2) = u_{w,\beta,2}(s_2, s_1) = A_{w,\beta,s_1,s_2}$, where the matrix $A_{w,\beta} \in \mathbb{R}^{2 \times 2}$ is given by $A_{w,\beta} = \begin{pmatrix} 1 & 0 \\ \beta & w \end{pmatrix}$, for $\beta \leq 1 \leq w$. The game $\Gamma_{w,\beta}$ has the same NE as the original game, $\Gamma$, retains its payoff- and risk-dominance properties, and preserves the limiting behavior of any QRD in $\Gamma$. Accordingly, each game $\Gamma_{w,\beta}$ has three NE, two pure at $x = y = 0$ and $x = y = 1$, with social welfare $\text{SW}(0,0) = 2w$ and $\text{SW}(1,1) = 2$, respectively, as well as one fully-mixed NE at $x^* = y^* = \alpha := w/(w+1-\beta)$.*

We are going to refer to the first pure-NE of $\Gamma_w$ as $x_w$. Note that $x_w$ is payoff-dominant for any $\Gamma_{w,\beta}$, and it is also risk-dominant whenever $w > 1 - \beta$, or equivalently, whenever $\alpha > 0.5$.

**Replicator dynamics (RD) versus gradient descent (GD).** The first result of this section is that whenever the risk- and payoff-dominant equilibria of $\Gamma_{w,\beta}$ coincide, i.e., $\alpha \geq 0.5$, then the gradient descent (GD) dynamics, i.e., the $0$-replicator dynamics, perform better (or equally in the generic case $\alpha = 0.5$) on average than the standard replicator dynamics (RD) with respect to the social welfare of their outcomes, i.e., they yield a smaller APoA. In any other instance of these games, i.e., for $\alpha < 0.5$, the RD perform better than GD with respect to the same metric.

**Theorem 4.6** ( Comparative Performance of RD and GD in $2 \times 2$ PRPGs)**.** *Consider an arbitrary $2 \times 2$ symmetric PRPG, which, without any loss of generality, can be represented as an instance $\Gamma_{w,\beta}$, and let $V_0, V_1$ denote the equations of motion of the $0$-replicator, i.e., gradient descent dynamics, and $1$-replicator, i.e., replicator dynamics, respectively, cf. equation QRD. Then, it holds that*

$$\text{APM}_{\text{SW,int}\,\mathcal{X}}(V_0, \Gamma_{w,\beta}) \geq \text{APM}_{\text{SW,int}\,\mathcal{X}}(V_1, \Gamma_{w,\beta}), \tag{1}$$

*if and only if the payoff-dominant equilibrium is also risk-dominant. Equality obtains only when $\alpha = 0.5$, i.e., when $w = 1 - \beta$.*

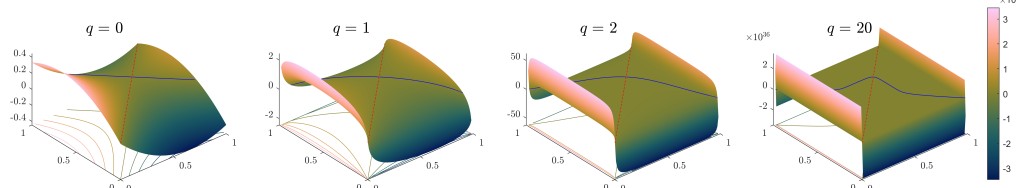

**Figure 1:** The invariant function, $\Psi_q(x, y)$, for all $x, y \in [0, 1]^2$ (cf. Lemma 4.7) in the game $\Gamma_{w,\beta}$ for $w = 2$, $\beta = 0$, and $q = 0$ (gradient descent), $q = 1$ (standard replicator), $q = 2$ (log-barrier), and $q = 20$. The invariant function becomes steep at the boundary as $q$ increases, taking both arbitrarily large negative (**dark**) and positive (light) values in the vicinity of the NE. The contour lines at zero, highlighted as solid blue and dashed red lines on the surfaces, correspond to the stable, $\Psi_{q,\text{Stable}}(x, y) = 0$, and unstable, $x - y = 0$, manifolds, respectively.

Theorem 4.6 demonstrates that the *riskiness* of the socially optimal equilibrium is decisive in characterising the relative performance of the GD and RD. In particular, as stated in Theorem 4.6, the expected social welfare is optimized by GD whenever risk and payoff-dominant equilibria coincide and is optimized by RD when risk and payoff-dominant equilibria differ. From a practical perspective, Theorem 4.6 provides a concrete recommendation on the optimal behavior of the agents (GD versus RD) based solely on the properties of the underlying game. However, it also suggests that even in low-dimensional, $2 \times 2$ potential games, there is not a uniform recommendation, and the optimal behavior largely depends on the features of the underlying game. As it turns out, in this case, the decisive feature is the *riskiness* of the payoff-dominant equilibrium (see also Appendix C).

**Invariant functions.** Technically, the proof of Theorem 4.6 (see Appendix C) proceeds with a first order analysis of the manifolds that separate the regions of attractions of the two pure equilibria for the different dynamics. This approach leverages the *constants of motion* or *invariant functions* (Nagarajan et al., 2020), i.e., quantities that remain constant along the trajectories of the learning dynamics. The rationale is that if we can identify such a function, then, by finding its value at the unique mixed equilibrium $\alpha$ of the game, we can determine all initial conditions that asymptotically converge to it: these will be all points at the same level set of the invariant function. The manifold, i.e., the geometric locus of all the points that converge to the equilibrium, formally, the *stable manifold* of $\alpha$, is the one that separates the regions of attractions of the two pure NE of the game. Because of this property, we may also refer to the stable manifold of the mixed NE as the *separatrix* (Panageas & Piliouras, 2016). Note that, since the dynamics are also backward-invariant (Panageas & Piliouras, 2016; Mertikopoulos & Sandholm, 2018), their level-set will also contain a set of initial conditions that converge to it when moving backward in time. These points constitute the *unstable manifold* of $\alpha$. In Lemma 4.7, we identify such an invariant for all QRD.

**Lemma 4.7** (Invariant functions of QRD in $2 \times 2$ symmetric PRPGs)**.** *Given a $2 \times 2$ symmetric PRPG, $\Gamma_{w,\beta}$, whose agents evolve with respect to the q-replicator dynamics, QRD, the separable function $\Psi_q : (0, 1)^2 \to \mathbb{R}$, with $\Psi_q(x, y) := \psi_q(x) - \psi_q(y)$, and $\psi_q : (0, 1) \to \mathbb{R}$ given by*

$$\psi_q(x) = \begin{cases} \dfrac{x^{2-q} + (1-x)^{2-q} - 1}{2-q} + \dfrac{1 - \alpha x^{1-q} - (1-\alpha)(1-x)^{1-q}}{1-q}, & q \neq 1, 2, \\ \alpha \ln(x) + (1-\alpha) \ln(1-x), & q = 1, \\ \ln(x) + \ln(1-x) + \dfrac{\alpha}{x} + \dfrac{1-\alpha}{1-x}, & q = 2, \end{cases} \quad (2)$$

*remains constant along any trajectory $\{x(t), y(t)\}_{t \geq 0}$ of the system. The function $\Psi_q(x)$ is continuous with respect to the parameter q at, both, $q = 1$ and $q = 2$, since $\lim_{q \to 1} \Psi_q(x) = \Psi_1(x)$ and $\lim_{q \to 2} \Psi_q(x) = \Psi_2(x)$ for all $x \in (0, 1)$.*

**Monotonicity of RoAs for all QRD.** A rigorous extension of Theorem 4.6 to all QRD is prevented by the difficulty to obtain an analytical form of the separatrix for all $q \in (0, 1)$. However, by visualizing the RoAs for all QRD in this range, we obtain empirical evidence that the RoA is monotonically increasing in $q$.

In Figure 1, we visualize the invariant function, $\Psi_q(x, y)$, for $x, y \in (0, 1)^2$ defined in Lemma 4.7 for various values of $q \in [0, 20]$. From the panels of Figure 1, it is also evident that $\Psi_q(x, y)$ acts as a handy tool to visualize the regions of attraction of the two pure NE of the game. Namely, at the unique

mixed NE, i.e., at $x = y = \alpha$, the invariant function, $\Psi_q$, is equal to 0. The same holds for any point $(x, y) \in (0, 1)^2$ with $x = y$. Thus, we can factorize $\Psi_q(x, y)$ as $\Psi_q(x, y) = \Psi_{q,\text{Stable}}(x, y) \cdot (x - y)$ where $\Psi_{q,\text{Stable}}(x, y) = 0$ is precisely the geometric locus of all points $(x, y) \in (0, 1)^2$ such that $\lim_{t \to \infty} x(t) = \alpha$, and $y = x$ is the geometric locus of all points such $\lim_{t \to -\infty} x(t) = \alpha$. These two manifolds constitute the *stable* and *unstable* manifolds of the $q$-replicator dynamics.

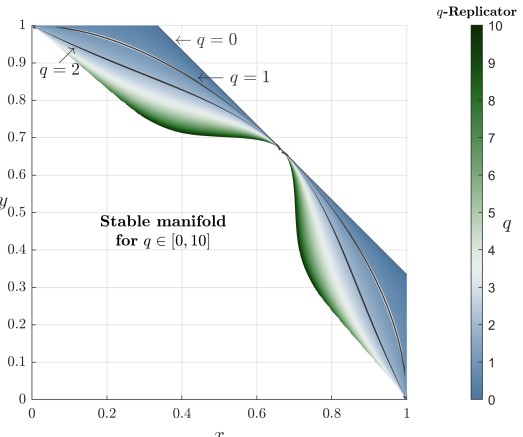

If we stack the stable manifolds (solid blue lines in Figure 1), it becomes evident that the region of attraction of the payoff-dominant and risk-dominant equilibrium grows as $q$ decreases to 0. This is depicted in Figure 2 for all values of $q \in [0, 10]$. In this case, the unstable manifolds are always equal to the diagonal, $x = y$ and thus, omitted. The progression of the surface of the stable manifolds continuous for larger values of $q$. Analogous plots, but with the results reversed as predicted by Theorem 4.6, can be generated for instances of $\Gamma_{w,\beta}$, in which the risk-dominant equilibrium is different from the payoff-dominant one, as well as, for $2 \times 2$ generic PRPGs. In general, putting together Theorem 4.6 and the current visualizations, we have both theoretical and empirical evidence that the region of attraction of the payoff-dominant equilibrium in $\Gamma_{w,\beta}$ is decreasing (increasing) in $q$ for $q \geq 0$ whenever this equilibrium is (is not) risk-dominant.

**Figure 2:** Stable manifold (separatrix) for all different values of $q \in [0, 10]$ (from blue to brown) in the $\Gamma_{w,\beta}$ game for $w = 2$ and $\beta = 0$. The manifolds for $q = 0$, $q = 1$, and $q = 2$ are shown in shades of black for reference (cf. Figure 5). The region of attraction of the payoff-dominant equilibrium (bottom-left corner) shrinks as $q$ increases.

**Application: APoA in $2 \times 2$ PRPGs.** To show-case the practical importance of Theorem 4.6 and the invariant function approach, we conclude this section with a concrete evaluation of the APoA average-performance measure in the class of $2 \times 2$ symmetric PRPGs. For this result, we focus on symmetric $2 \times 2$ PRPGs such that payoff and risk-dominant equilibria coincide; one can prove tight bounds on APoA as shown in Theorem 4.8.

**Theorem 4.8.** *The APoA of GD dynamics in all $2 \times 2$ symmetric PRPGs, $\Gamma_{w,\beta}$, in which the payoff- and risk-dominant NE coincide is bounded by 2, i.e., $APoA(V_0, \Gamma_{w,\beta}) \leq 2$, for all $\Gamma_{w,\beta}$ with $\beta > 1 - w$. This bound is tight.*

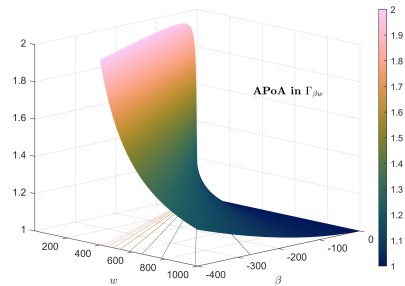

**Figure 3:** APoA of a $2 \times 2$ symmetric PRPG for the gradient descent dynamics and various values of $\beta$ and $w$. The APoA is upper bounded by 2 (**dark** to light values) as shown in Theorem 4.8. similar bounds hold for all values of $q$.[5]

The proof of Theorem 4.8 proceeds by first order analysis of the function depicted in Figure 3 which, in turn, depends on the invariant function of the gradient descent dynamic, see Appendix C. Note that the bound of Theorem 4.8 also holds for $\beta = 1 - w$, but in this case, there exists no risk-dominant equilibrium. Moreover, one way to check that this bound is tight, is to set $\beta = 1 - w + \epsilon$, for a small $\epsilon > 0$ and let $w$ increase (cf. Figure 3). In combination Theorem 4.6 and Theorem 4.8 imply that the APoA of the RD (QRD with $q = 1$), is *not* upper bounded by 2 whenever $\alpha < 0.5$, i.e., whenever the risk- and payoff-dominant equilibria are different. However, for the case $\alpha > 0.5$, the separatrices for all $q \geq 0$ as visualized in Figure 2, (empirically) imply that

---

[5]In Figure 2, we visualize the stable manifolds when GD have the *largest* region of attraction, i.e., the lowest APoA. The case $\alpha < 0.5$, in which the manifolds are mirrored on the $y = 1 - x$ diagonal, is in Appendix C.

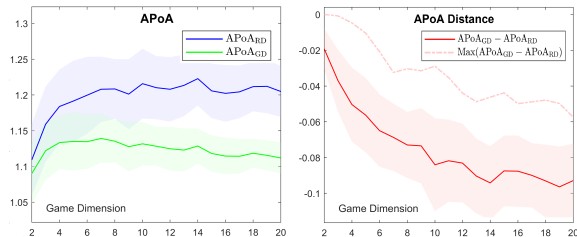

| | APoA $\pm$ std (Max APoA) | |
|---|---|---|
| m | RD | GD |
| 3 | $1.16 \pm .05(1.23)$ | $1.12 \pm .04(1.19)$ |
| 5 | $1.19 \pm .06(1.30)$ | $1.14 \pm .04(1.22)$ |
| 10 | $1.22 \pm .05(1.38)$ | $1.13 \pm .03(1.20)$ |
| 20 | $1.21 \pm .04(1.28)$ | $1.11 \pm .02(1.22)$ |

**Figure 4 & Table 1:** Numerical results for the APoA metric of replicator dynamics (RD) and gradient descent (GD) in diagonal PRPGs of dimensions $m = 2, \ldots, 20$. The two panels show the APoA of each dynamic together with its standard deviation (left), their difference (solid line enveloped by the standard deviation shaded region) and the difference of the maximum APoAs of the dynamics (right) observed over 100 sampled games at each dimension. The GD dynamics throughout outperform the RD dynamics suggesting that Theorem 4.6, with the notions of pure- and risk-dominant NE properly refined, may still hold for larger dimensional PRPGs (for the reverse implication in Theorem 4.6 see Section 5). Table 1 provides detailed statistics for $m = 3, 5, 10$ and $20$.

## 5 HIGHER DIMENSIONAL NUMERICAL EXPERIMENTS

In this section, we present experimental evidence that our approach can be extrapolated on larger setups, i.e., beyond $2 \times 2$-games. Notice that, the extrapolation needs to be done with care since the central notion of risk dominance may not be well-defined in larger game setups. We choose to treat this, by restricting our focus to the class of two-player diagonal PRPGs. Such games can be described by a common payoff matrix $U$ the only non-zero elements of which lie on its diagonal, i.e., it is of the form, $U := \mathrm{diag}(u_1, u_2, \ldots, u_m)$, where w.l.o.g, we assume $0 < u_1 < u_2 < \cdots < u_m$. Notice that a diagonal PRPG defined as above has the following important traits:

1. Profiles $(1, 1), \ldots, (m, m)$ are the only pure NE and, since the game is PRPG, the only stable NE.
2. The pure NE can be ordered in terms of payoffs as $u_1 < \cdots < u_m$. Consequently the NE $(i, i)$ dominates all NE $(j, j)$, $j = 1, \ldots, i - 1$.
3. The difference in payoffs from any pure NE $(i, i)$ to some non-diagonal profile $(i, j)$, or $(j, i)$, is $u_i$ for any $j \neq i$. If we define risk to be the negative deviation payoffs Harsanyi & Selten (1988), then the risk of NE $(i, i)$ is $-u_i^2$ for all $i = 1, \ldots, m$, and therefore the pure NE can be ordered in terms of risk with NE $(i, i)$ be less risky than $(j, j)$ for all $j = 1, \ldots, i - 1$.

**Experimental setup.** We run experiments in random 2-agent, symmetric diagonal PRPGs of dimensions $m = 2, 3, \ldots, 20$ (size of each agent's action space). In each game, the payoffs $u_1, u_2, \ldots, u_m$ are selected (pseudo-)randomly and satisfy the following properties: (i) the lowest diagonal payoff, $u_1$, is at least as large as some predefined positive constant (set equal to $1e - 12$ for the experiments), (ii) the highest payoff, $u_m$, is equal to the dimension, $m$, of the game, i.e., $u_m = 2, 3, \ldots$ and 20 respectively, and (iii) $u_2, \ldots, u_{m-1}$ are in ascending order strictly between $u_1$ and $u_m$ with randomly selected distances between them. For each dimension, we sample 100 random games and run the gradient descent and standard replicator dynamics for 1000 initial conditions till convergence.

The outputs of the simulations of the above experiments are summarized in Figure 4 and Table 1. The outputs provide indications that (i) the gradient descent dynamics (continue to) outperform the replicator dynamics in all diagonal games in terms of average performance, and that (ii) the APoA is lower-bounded by 2, the theoretical bound that we presented in the case of $2 \times 2$ games. As Table 1 suggests, the APoA is, in fact, considerably lower than 2 in all sampled games.

## 6 CONCLUSION

We study the class of $q$-replicator dynamics (QRD), and showed that all QRD converge pointwise to Nash equilibria in perfectly-regular potential games, a class of games that encompasses almost all potential games, i.e., the standard models of multi-agent coordination. From the perspective of equilibrium selection and quality, however, convergence provides little information. Even if two dynamics converge and even if they have joint optimal no-regret guarantees, they may still exhibit very different equilibrium selection properties, which, in turn, determine their practical performance. Our analysis provides the first formal comparative analysis of different optimization-driven dynamics by combining diverse techniques and establishes an intriguing direction for future work.

ACKNOWLEDGMENTS

This research was supported in part by the National Research Foundation, Singapore and DSO National Laboratories under its AI Singapore Program (AISG Award No: AISG2-RP-2020-016), grant PIESGP-AI-2020-01, AME Programmatic Fund (Grant No.A20H6b0151) from A*STAR and Provost's Chair Professorship grant RGEPPV2101. Ioannis Panageas would like to acknowledge startup grant from UCI. Part of this work was conducted while Ioannis was visiting Archimedes Research Unit. This work has been partially supported by project MIS 5154714 of the National Recovery and Resilience Plan Greece 2.0 funded by the European Union under the NextGenerationEU Program.

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

# Supplementary Material for

# Beating Price of Anarchy and Gradient Descent without Regret in Potential Games

## A    Extensions and challenges

A rigorous average-case analysis in arbitrary games is mathematically and computationally challenging, since closed-form solutions for the boundaries of RoAs not always exist, and since computing RoAs is generally NP-Hard. Nevertheless, it is immediate to extend the empirical analysis by approximating the APM of interest through counts of frequencies for each limiting equilibrium for multiple random initializations. However, the invariant function approach offers a scalable alternative that is promising to work in more complicated settings. Finally, non-uniform distributions over initial conditions that keep the expectation in Definition 4.3 well-defined can be used to study games in which the dynamics converge to limit cycles instead of NE.

## B    Missing Proofs and Materials: Section 3

To complete the steps in the sketch of the proof of Theorem 3.2, we start by showing that the class of perfectly-regular potential games (PRPG) is well-defined, since restrictions of potential games are also potential games.

**Lemma B.1.** *A restriction of a potential game is also a potential game.*

*Proof of Lemma B.1.* Let $\Gamma'$ a restriction of a potential game $\Gamma$ and potential function $\Phi : \mathcal{A} \to \mathbb{R}$. Take $\Phi' : \mathcal{A}' \to \mathbb{R}$ to be the restriction of $\Phi$ to $\mathcal{A}' \subseteq \mathcal{A}$. Then for all $k \in \mathcal{N}$, and $s, s' \in \mathcal{A}'$ we have that:

$$u'_k(s) - u'_k(s'_k, s_{-k}) = u_k(s) - u_k(s'_k, s_{-k}) = \Phi(s) - \Phi(s'_k, s_{-k}) = \Phi'(s) - \Phi'(s'_k, s_{-k}),$$

where the second equality follows from the definition of potential games (cf. section 2). Hence, $\Phi'$ is a potential function, and therefore $\Gamma'$ a potential game.    □

Using the recent results of Swenson et al. (2020), it is not difficult to show that perfectly-regular potential games are generic, and have a finite number of restricted equilibria. These are the statements of Lemma B.2 and Lemma B.3, respectively.

**Lemma B.2.** *Almost all finite potential games are perfectly-regular.*

*Proof of Lemma B.2.* Let RPG, and PRPG denote the sets of regular potential, and perfectly-regular potential games, respectively. Let also $\Gamma$ be a random finite potential game. Since $\Gamma$ is finite, there exist $2^m$ distinct restrictions of $\Gamma$, where $m := \sum_{k \in \mathcal{N}} |\mathcal{A}_k|$. Then, by Lemma B.1, we have that any restriction $\Gamma'$ of $\Gamma$ is also a random finite potential game, and therefore $\Pr(\Gamma' \in \text{RPG}) = 1$, with respect to the Lebesgue measure Swenson et al. (2020). It follows that:

$$\Pr(\Gamma \in \text{PRPG}) = 1 - \Pr(\Gamma \notin \text{PRPG})$$
$$= 1 - \Pr(\bigcup_{\Gamma' \in R_\Gamma} \Gamma' \notin \text{RPG})$$
$$\geq 1 - \sum_{\Gamma' \in R_\Gamma} \Pr(\Gamma' \notin \text{RPG}) = 1,$$

where the last equality follows from the fact that $|R_\Gamma|$ is finite.    □

**Lemma B.3.** *Every perfectly-regular finite potential game has a finite number of restricted equilibria.*

*Proof of Lemma B.3.* Let $\Gamma$ be a perfectly-regular finite potential game, and let $\Gamma'$ be one of its restrictions. By the definition of a perfectly-regular potential game, we have that $\Gamma'$ is a regular potential game. Furthermore, since $\Gamma$ is finite and $\mathcal{A}'_k \subseteq \mathcal{A}_k$ for all $k \in \mathcal{N}$, it follows that $\Gamma'$ is finite. But then $\Gamma'$ is a finite regular potential game and as such it has a finite number of Nash equilibria, i.e., $\text{NE}(\Gamma') < \infty$ Swenson et al. (2020). Finally, since each restricted equilibrium is a Nash equilibrium of a restrictions of $\Gamma$, it follows that there exist at most:

$$\sum_{\Gamma' \in R_\Gamma} |\text{NE}(\Gamma')| \leq 2^m \max_{\Gamma' \in R_\Gamma} |\text{NE}(\Gamma')| < \infty$$

restricted equilibria of $\Gamma$. Therefore, the number of restricted equilibria of $\Gamma$ is finite. □

We, now, consider the $q$-replicator dynamics of a finite potential game $\Gamma$, given by the dynamical system of equations in equation QRD. Our goal for the remainder of this section is to prove that for any interior initial condition $x(0) \in \mathcal{X}$ the dynamics in equation QRD converge pointwise to a Nash equilibrium of a given perfectly-regular finite potential game.[1]

The proof proceeds in two parts: First, we prove that the dynamics converge to a restricted equilibrium of the game for any initial condition. Second, we prove that for any *interior* initial condition, the dynamics are bound to deviate from any rest-point that is not a Nash equilibrium, and therefore, they have to converge to a Nash equilibrium.

Let us begin by proving the first of the two claims. For this, we will use the notion of the $\omega$-*limit set* of a sequence $(x(t))_{t \geq 0} \subseteq \mathcal{X}$ that is generated by the QRD, which is defined as

$$\omega(x(t)) := \bigcap_{t \geq 0} \text{cl}\{x(t') \mid t' > t\}$$

where $\text{cl}\, S$ denotes the closure of a set $S$.

**Lemma B.4.** *Given a perfectly-regular finite potential game $\Gamma$, every $\omega$-limit set, with respect to the $q$-replicator dynamics, is a singleton $\{x^*\}$, where $x^* \in \mathcal{X}$ is a rest-point of the dynamics. Specifically, $x^*$ is a Nash equilibrium, if $q = 0$, or a restricted equilibrium, if $q > 0$. Furthermore, the set $\mathcal{Q}(\mathcal{X}) := \bigcup_{x_0 \in \mathcal{X}}\{x^* \in \mathcal{X} \mid \lim_{t \to \infty} x(t) = x^*,\ x(0) = x_0\}$, i.e., the set of all limit points, is finite.*

*Proof of Lemma B.4.* Let $\Gamma$ be a perfectly-regular finite potential game. Since $\Gamma$ is a potential game, by Proposition 6.4 of Mertikopoulos & Sandholm (2018), we have that every $\omega$-limit set consists entirely of rest-points of the dynamics. In particular, these are Nash equilibria of $\Gamma$, if $q = 0$, or restricted equilibria of $\Gamma$, if $q > 0$. However, since $\Gamma$ is perfectly-regular—it suffices for it to be regular for the case of $q = 0$—it follows by Lemma B.3 that every $\omega$-limit set is a finite set. Consider now, the $\omega$-limit set of an orbit $(x(t))_{t \geq 0}$ of the dynamics for some arbitrary initial condition $x(0) = x_0$. Since $x(t)$ is continuous, the $\omega$-limit set $\omega(x(t))$ is the decreasing intersection of compact, connected sets and, therefore, it is connected. Since the $\omega$-limit set is finite, the above implies that, in fact it has to be a singleton $\{x^*\}$, where $x^*$ is a Nash equilibrium if $q = 0$ Mertikopoulos & Sandholm (2018), or a restricted equilibrium if $q > 0$, respectively. Finally, from the above, we have that the set of all limit points, $\mathcal{Q}(\mathcal{X})$ is a subset of the restricted equilibria of $\Gamma$; therefore, since $\Gamma$ is a perfectly-regular finite potential game, we have, by Lemma B.3, that the set of restricted equilibria of $\Gamma$ and, consequently, $\mathcal{Q}(\mathcal{X})$ are finite. □

To prove Theorem 3.2, it remains to exclude convergence to restricted equilibria that are not NE of the original game, $\Gamma$. To establish that, we will show that as the QRD approach a limit point $x^*_k$, the probability $x_{k,a_k}$ of non-optimally performing actions must go to zero. Thus, all actions in $\text{supp}\, x^*_k$ must be a best response against $x^*_{-k}$ for all agents $k \in \mathcal{N}$ which implies that $x^*_k$ is a NE of $\Gamma$.

*Proof of Theorem 3.2.* If $q = 0$, the statement follows directly from Lemma B.4. So we only need to consider the $q$-Replicator Dynamics for $q > 0$. Let $\Gamma$ be a perfectly-regular finite potential game, and let $(x(t))_{t \geq 0}$ be a trajectory of the $q$-replicator dynamics with initial condition $x(0) = x_0 \in \text{int}\, \mathcal{X}$.

---

[1] Recall that the *interior* of the set $\mathcal{X}$, $\text{int}\, \mathcal{X}$ is the set of all joint choice distributions $x \in \mathcal{X}$ with full support, i.e., $x_{k,a_k} > 0$ for all $a_k \in \mathcal{A}_k$ and for all $k \in \mathcal{N}$; all points of $\mathcal{X}$ that are not in the interior, are called *boundary points*.

Since $q > 0$, we know that the support of $x(t)$ remains constant for all $t \in \mathbb{R}$ Mertikopoulos & Sandholm (2018). Thus, since $x(0) \in \text{int } \mathcal{X}$, it follows that $x(t)$ remains in the interior of $\mathcal{X}$ for all $t \geq 0$, i.e., $x_{k,a_k}(t) > 0$ for all $k \in \mathcal{N}$ and for all $a_k \in \mathcal{A}_k$. Furthermore, by Lemma B.4, we have that the limit $\lim_{t \to \infty} x(t)$ exists and is a rest-point of the dynamics, say $x^*$.

Assume, now, that $x^*$ is not a Nash equilibrium. This implies that there exists a player $i$ and a pure action $a_i$ of $i$ such that $u_i(a_i, x^*_{-i}) > u_i(x^*)$. Moreover, since, $x^*$ is, by construction, a rest-point, $\dot{x}^*_{ia_j} = 0$ for all actions $a_j$. Observe that $x^*_{ia_i} = 0$. To see this, let us assume the absurdum $x^*_{ia_i} > 0$. Since $\dot{x}^*_{ia_j} = 0$ for all $a_j$, by equation QRD, we have

$$u_i(a_j, x^*_{-i}) = \frac{\sum_{a_k \in \mathcal{A}_i} {x^*_{ia_k}}^q u_i(a_k, x^*_{-i})}{\sum_{a_k \in \mathcal{A}_i} {x^*_{ia_k}}^q}, \quad \text{for all } a_j \text{ whenever } x_{ia_j} > 0. \tag{3}$$

Furthermore, since $u_i(a_i, x^*_{-i}) > u_i(x^*)$, we also have

$$\begin{aligned}
u_i(a_i, x^*_{-i}) &> u_i(x^*) \\
&= \sum_{a_j \in \mathcal{A}_i} x^*_{ia_j} u_i(a_j, x^*_{-i}) \\
&= \sum_{a_j \in \mathcal{A}_i} x^*_{ia_j} \frac{\sum_{a_k \in \mathcal{A}_i} {x^*_{ia_k}}^q u_i(a_k, x^*_{-i})}{\sum_{a_k \in \mathcal{A}} {x^*_{ia_k}}^q} \quad \text{by equation 3} \\
&= \frac{\sum_{a_k \in \mathcal{A}_i} {x^*_{ia_k}}^q u_i(a_k, x^*_{-i})}{\sum_{a_k \in \mathcal{A}_i} {x^*_{ia_k}}^q}.
\end{aligned} \tag{4}$$

Clearly, the above reveals a contradiction, and therefore, it must be the case that $x^*_{ia_i} = 0$.

Fix $\epsilon > 0$, and consider the set:

$$B_\epsilon := \left\{ x \in \mathcal{X} \mid u_i(a_i, x_{-i}) > \frac{\sum_{a_j \in \mathcal{A}_i} x^q_{i,a_j} u_i(a_j, x_{-i})}{\sum_{a_j \in \mathcal{A}_i} x^q_{i,a_j}} + \epsilon \right\}.$$

By continuity, $B_\epsilon$ is *open* and by equation equation 4, it contains $x^*$—given $\epsilon$ small enough. Since $x(t)$ converges to $x^*$ as $t \to \infty$, there exists a time $t_\epsilon \geq 0$ such that $x(t) \in B_\epsilon$ for all $t > t_\epsilon$. Therefore, for each $t > t_\epsilon$ we have that:

$$\dot{x}_{i,a_i}(t) = (x_{i,a_i}(t))^q \left( u_i(a_i, x_{-i}) - \frac{\sum_{a_j \in \mathcal{A}_i} (x_{i,a_j}(t))^q u_i(a_j, x_{-i})}{\sum_{a_j \in \mathcal{A}_i} (x_{i,a_j}(t))^q} \right) > \epsilon (x_{i,a_i}(t))^q > 0,$$

where the last inequality follows because $x(t) \in \text{int } \mathcal{X}$ for all $t \geq 0$. Finally, by integrating with respect to time, we have that for all $t > t_\epsilon$:

$$x_{i,a_i}(t) = \int_{t'=t_\epsilon}^{t} \dot{x}_{i,a_i}(t') \, dt + x_{i,a_i}(t_\epsilon) > x_{i,a_i}(t_\epsilon) > 0.$$

Therefore, by the continuity of $x(t)$, we have that $x^*_{ia_i} = \lim_{t \to \infty} x_{i,a_i}(t) \geq x_{i,a_k}(t_\epsilon) > 0$, which is a contradiction to our assumption that $x^*_{ia_i} = 0$, which is a direct consequence of $x^*$ *not* being a NE of the game; thus, $x^*$ has to be a Nash equilibrium of $\Gamma$. $\qquad \square$

## C   MISSING PROOFS AND MATERIALS: SECTION 4

For this part, it is useful to use the generic notation used by Pangallo et al. (2022), who provide a taxonomy of $2 \times 2$ games. Consider an arbitrary $2 \times 2$ symmetric PRPG with payoff with payoff functions $u_1(s_1, s_2) = u_2(s_2, s_1) = B_{s_1, s_2}$, where the payoff matrix $B \in \mathbb{R}^{2 \times 2}$ is given by:

$$B = \begin{pmatrix} a & c \\ b & d \end{pmatrix}, \tag{5}$$

and set without any loss of the generality $d \geq a \geq b$—that may always be done by possibly re-indexing the agents' actions. If $c > d$, then the game is dominance-solvable (cf. Table 1 of Pangallo et al. (2022)) and as such the dynamics of the game are trivial. Hence, we may narrow our scope

to payoff matrices that satisfy $d \geq c$. Also, PRPG games have a finite number of Nash equilibria, and so, we may exclude games where the above inequalities are *not strict*. All in all, we are going to assume without any loss of the generality that the following conditions hold:

$$d > a > b \quad \text{and} \quad d > c. \tag{6}$$

*Proof of Lemma 4.5.* Recall that in the notation of Pangallo et al. (2022), an arbitrary $2 \times 2$ symmetric PRPG can be written with payoff functions $u_1(s_1, s_2) = u_2(s_2, s_1) = B_{s_1, s_2}$, where the payoff matrix $B \in \mathbb{R}^{2 \times 2}$ is given by $B = \begin{pmatrix} a & c \\ b & d \end{pmatrix}$, where we assumed without loss of generality that $d > a > b$ and $d > c$ (cf. equation equation 6. Let also $(x, 1-x)$ and $(y, 1-y)$ with $x, y \in [0, 1]$ denote the choice distributions of the two agents, adopting the common notation for the statespace of game dynamics in $2 \times 2$ games. Thus, by slightly abusing notation, the choice distributions can be conveniently represented by single variables, $x$ and $y$ for agents 1 and 2, respectively. The 3 Nash equilibria of such games are $x = y = 1$ and $x = y = 0$, with payoffs $a$ and $d$, respectively, for both players, and $x^* = y^* = \frac{d-c}{a-b+d-c}$, with payoff $(x^*)^2 a + x^*(1-x^*)(b+c) + (1-x^*)^2 d$ for both players. Recall that, by definition, the Nash equilibrium $x = y = 0$ is always *payoff-dominant*—due to the possible re-indexing of the actions—and it is *risk-dominant* if $d - c > a - b$.

To prove the claim of the Lemma, we begin by presenting the equations of motion of the $q$-replicator dynamics as functions of $x$ and $y$. For the first agent that is:

$$\dot{x} = x^q \left( u_1(a_1, y) - \frac{x^q u_1(a_1, y) + (1-x)^q u_1(a_2, y)}{x^q + (1-x)^q} \right) \tag{7}$$

$$= \frac{x^q (1-x)^q}{x^q + (1-x)^q} (u_1(a_1, y) - u_1(a_2, y))$$

$$= \frac{x^q (1-x)^q}{x^q + (1-x)^q} (ay + c(1-y) - by - d(1-y)) \tag{8}$$

$$= \frac{x^q (1-x)^q}{x^q + (1-x)^q} [(a - b + d - c)y - (d - c)]$$

$$= \frac{x^q (1-x)^q}{x^q + (1-x)^q} \kappa \cdot (y - y^*), \tag{9}$$

where $\kappa := a - b + d - c$. Similarly, we may derive the equation of motions for the second agent as $\dot{y} = \frac{y^q (1-y)^q}{y^q + (1-y)^q} \kappa \cdot (x - x^*)$. Here, $(x^*, y^*)$ is the mixed Nash equilibrium of the game (see above), and it holds that $y^* = x^*$. Thus, apart from the variables $x$ and $y$ and the hyperparameter $q$ which is exogenously given, the $q$-replicator dynamics depend on the payoffs of the game $\Gamma$ only through $\kappa$ and $x^* = \frac{d-b}{\kappa}$. It follows that any transformation that preserves the value of $x^*$ and scales $\kappa$ by a constant may only scale $\dot{x}$ and $\dot{y}$ by the same constant; that is, may only affect the convergence rate of the dyanmics, but *not* their limiting behavior. Starting from an arbitrary payoff matrix $B$ as given in equation 5 let us assume, without any loss of the generality, that the conditions in equation 6 apply. Notice that, since by the aforementioned assumptions we have that $d > c$, we may set some $\delta \in \mathbb{R}$ such that $0 < a + \delta < d - c$. Accordingly, we consider the following sequence of transformations: (T1) Add $\delta$ to the first column, (T2) subtract $c$ from the second column, and (T3) divide by $a + \delta$. These lead to:

$$B = \begin{pmatrix} a & c \\ b & d \end{pmatrix} \xrightarrow{\text{(T1)}} \begin{pmatrix} a + \delta & c \\ b + \delta & d \end{pmatrix} \xrightarrow{\text{(T2)}} \begin{pmatrix} a + \delta & 0 \\ b + \delta & d - c \end{pmatrix} \xrightarrow{\text{(T3)}} \begin{pmatrix} 1 & 0 \\ \frac{b+\delta}{a+\delta} & \frac{d-c}{a+\delta} \end{pmatrix} =: A_{\frac{d-c}{a+\delta}, \frac{b+\delta}{a+\delta}}.$$

Notice that $A_{\frac{d-c}{a+\delta}, \frac{b+\delta}{a+\delta}}$ is the payoff matrix of a parametric game $\Gamma_{w, \beta}$, where $w := \frac{d-c}{a+\delta}$ and $\beta := \frac{b+\delta}{a+\delta}$. Observe, that (T1), (T2) and (T3) leave $x^*$ unaltered and only scale $\kappa$ by a constant $\frac{1}{a+\delta}$; that is, the limiting behavior of the $q$-replicator dynamics is preserved. Furthermore, the risk-dominance of the equilibrium points is preserved, because:

$$d - c > a - b \quad \text{if and only if} \quad \frac{d-c}{a+\delta} > 1 - \frac{b+\delta}{a+\delta}$$

Finally, the payoff-dominance of the Nash equilibrium $x = y = 0$ is also preserved because, by the definition of $\delta$, we have that $\frac{d-c}{a+\delta} > 1$. $\qquad \square$

Next, we are going to construct the invariant functions of $\Gamma_{w,\beta}$ with respect to the $q$-replicator dynamics, which are given by Lemma 4.7 that we restate below for completeness. Recall that $(\alpha, \alpha)$ is defined to be the equilibrium point of the a game $\Gamma_{w,\beta}$; that is, $x^* = y^* = \alpha$.

**Lemma 4.7** (Invariant functions of QRD in $2 \times 2$ symmetric PRPGs). *Given a $2 \times 2$ symmetric PRPG, $\Gamma_{w,\beta}$, whose agents evolve with respect to the $q$-replicator dynamics, QRD, the separable function $\Psi_q : (0,1)^2 \to \mathbb{R}$, with $\Psi_q(x,y) := \psi_q(x) - \psi_q(y)$, and $\psi_q : (0,1) \to \mathbb{R}$ given by*

$$\psi_q(x) = \begin{cases} \dfrac{x^{2-q} + (1-x)^{2-q} - 1}{2-q} + \dfrac{1 - \alpha x^{1-q} - (1-\alpha)(1-x)^{1-q}}{1-q}, & q \neq 1, 2, \\[2mm] \alpha \ln(x) + (1-\alpha) \ln(1-x), & q = 1, \\[2mm] \ln(x) + \ln(1-x) + \dfrac{\alpha}{x} + \dfrac{1-\alpha}{1-x}, & q = 2, \end{cases} \qquad (2)$$

*remains constant along any trajectory $\{x(t), y(t)\}_{t \geq 0}$ of the system. The function $\Psi_q(x)$ is continuous with respect to the parameter $q$ at, both, $q = 1$ and $q = 2$, since $\lim_{q \to 1} \Psi_q(x) = \Psi_1(x)$ and $\lim_{q \to 2} \Psi_q(x) = \Psi_2(x)$ for all $x \in (0,1)$.*

*Proof of Lemma 4.7.* To prove the statement, we will show that the time derivative of the function $\Psi_q(x(t), y(t))$ is equal to zero. Let us begin by constructing the derivative of $\psi(x)$. For $q \neq 1, 2$ we have that:

$$\psi_q'(x) = \frac{x}{x^q} - \frac{1-x}{(1-x)^q} - \frac{\alpha}{x^q} + \frac{1-\alpha}{(1-x)^q} = \frac{x-\alpha}{x^q} + \frac{x-\alpha}{(1-x)^q} = \frac{(x-\alpha)[(1-x)^q + x^q]}{x^q(1-x)^q}.$$

Similarly, for $q = 1$ we have that:

$$\psi_1'(x) = \frac{\alpha}{x} - \frac{1-\alpha}{1-x} = \frac{\alpha - x}{x(1-x)},$$

and, for $q = 2$, we have that:

$$\psi_2'(x) = \frac{1}{x} - \frac{1}{1-x} - \frac{\alpha}{x^2} + \frac{1-\alpha}{(1-x)^2} = \frac{(x-\alpha)[x^2 + (1-x)^2]}{x^2(1-x)^2}.$$

That is the derivative of $\psi(x)$ has the general form:

$$\psi_q'(x) = \lambda \cdot \frac{(x-\alpha)[(1-x)^q + x^q]}{x^q(1-x)^q},$$

for all $q \geq 0$, where $\lambda \in \{1, -1\}$. Notice, that the choice for $\lambda$ is purely stylistic because the invariance of a function is not affected by scalar transformations. Using equation 7 from the proof of Lemma 4.5 we have that:

$$\begin{aligned} \dot{\Psi}_q(x,y) &= \frac{\partial \Psi_q(x,y)}{\partial x} \dot{x} - \frac{\partial \Psi_q(x,y)}{\partial y} \dot{y} \\ &= \psi_q'(x)\dot{x} - \psi_q'(y)\dot{y} \\ &= \lambda \kappa [(x-\alpha)(y-\alpha) - (y-\alpha)(x-\alpha)]. \end{aligned} \qquad \square$$

To proceed with the proof of Theorem 4.6, we first need to provide the formal definition of the stable and unstable manifolds of the mixed NE of $\Gamma_{w,\beta}$.

**Definition C.1** (Stable and unstable manifolds of $\Gamma_{w,\beta}$ under QRD). *Let $\Psi_q : [0,1]^2 \to \mathbb{R}$ with $\Psi_q(x,y) = \Psi_{q,\text{Stable}}(x,y) \cdot (x-y)$ for all $x, y \in [0,1]$ denote the invariant function of the $q$-replicator dynamics for the $2 \times 2$ symmetric PRPG, $\Gamma_{w,\beta}$. The* unstable manifold *of the mixed NE $(x,y) = (\alpha, \alpha)$ under the $q$-replicator dynamics is the curve $\mathcal{M}_{\text{Unstable}} := \{(x,y) \in (0,1)^2 \mid x = y\}$; that is, the set of points for which $\lim_{t \to -\infty} x(t) = \lim_{t \to -\infty} y(t) = \alpha$. Analogously, the* stable manifold *of the mixed NE is the curve $\mathcal{M}_{\text{Stable}} := \{(x,y) \in (0,1)^2 \mid \Psi_{q,\text{Stable}}(x,y) = 0\}$; that is, the set of points for which $\lim_{t \to \infty} x(t) = \lim_{t \to \infty} y(t) = \alpha$.*

We are now ready to prove Theorem 4.6.

*Proof of Theorem 4.6.* To establish the claim, We will need to have at hand an explicit form of the stable manifold $\mathcal{M}_{\text{Stable}}$ of the mixed NE of $\Gamma_{w,\beta}$ with respect to the 0-replicator dynamics, and GD. In that regard, we are now going to construct that manifold. Recall that from equation 2, the invariant function of any QRD in $\Gamma_{w,\beta}$ is given by $\Psi_0(x, y) := \psi_0(x) - \psi_0(y) = 0$, where:

$$\psi_0(x) = \frac{x^2 + (1-x)^2 - 1}{2} + 1 - \alpha x - (1-\alpha)(1-x) = x^2 - 2\alpha x + \alpha.$$

Therefore, $\Psi_0(x, y) = x^2 - 2\alpha x + \alpha - y^2 + 2\alpha y - \alpha = (x - y)(x + y - 2\alpha) = 0$ and the stable manifold (cf. subsection 4.2) satisfies $\Psi_{q,\text{Stable}}(x, y) = x + y - 2\alpha = 0$. In other words, the stable manifold is the line segment:

$$\mathcal{M}_{\text{Stable}} = \{(x, y) \in (0, 1)^2 \mid y = 2\alpha - x\}. \tag{10}$$

By equation 10 and Definition C.1, the stable and unstable manifolds of gradient descent (GD) are given by the lines $y = 2\alpha - x$ and $y = x$, respectively, where $\alpha = \frac{w}{w+1-\beta}$, while the stable and unstable manifolds of the standard replicator dynamics are given as solutions to:

$$\Psi_1(x, y) = \psi_1(x) - \psi_1(y) = \alpha \ln\left(\frac{x}{y}\right) + (1-\alpha) \ln\left(\frac{1-x}{1-y}\right) = 0,$$

where the line $y = x$ corresponds to a solution. We are going to prove that the *single* remaining solution of the previous equation, although it cannot be expressed explicitly, satisfies $y \leq 2\alpha - x$, if $\alpha > \frac{1}{2}$, $y \geq 2\alpha - x$, if $\alpha < \frac{1}{2}$ (with equality in both cases only if $x = \alpha$), and $y = 2\alpha - x$ otherwise; hence, the statements of Theorem 4.6 follow naturally.

It is not difficult to show that $\psi_1'(x) = \frac{\alpha - x}{x(1-x)}$ (cf. proof of Lemma 4.7), and $\psi_1''(x) = -\frac{x^2 - 2\alpha x + \alpha}{x^2(1-x)^2} < 0$. Therefore, $\psi_1$ is a strictly concave function with maximum at $x = \alpha$. To proceed, it will be useful to define the implicit function $y : (0, 1) \to (0, 1)$ such that $y(\alpha) = \alpha$, and $\forall x \in (0, 1) \setminus \{\alpha\}$: $y(x) \neq \alpha$, and $\psi_1(y(x)) = \psi_1(x)$. By applying the Intermediate Value Theorem (IVT) on $\psi_1$ in the intervals $(0, \alpha)$, and $(\alpha, 1)$, we can verify that $y$ is a well-defined bijective function. Note that $y = y(x)$ has to correspond to the remaining solution of $\Psi_1(x, y)$.

Without any loss of the generality, let us consider the case of $\alpha > \frac{1}{2}$. Since $\psi_1'$ is strictly decreasing in $(0, 1)$ ($\psi_1'' < 0$), we have that $\psi_1'(x) > 0$ for all $x \in (0, \alpha)$, and $\psi_1'(x) < 0$ for all $x \in (\alpha, 1)$. We begin by proving that for all $x \in (0, 1 - \alpha)$ it holds that $|\psi_1'(\alpha - x)| < |\psi_1'(\alpha + x)|$, i.e., $\psi_1'(\alpha - x) < -\psi_1'(\alpha + x)$. Specifically, we have the following series of equivalences:

$$\psi_1'(\alpha - x) < -\psi_1'(\alpha + x) \iff \frac{x}{(\alpha - x)(1 - \alpha + x)} < \frac{x}{(\alpha - x)(1 - \alpha - x)}$$

Since $\alpha > 1/2$ and $x \in (0, 1 - \alpha$, the latter is equivalent to $2x^2(1 - 2\alpha) < 0$ or equivalently to $\alpha > 1/2$ which holds by assumption. Next, by taking advantage of the above, we can prove that, for all $x \in (0, 1 - \alpha)$, it holds $\psi_1(\alpha - x) > \psi_1(\alpha + x)$; that is:

$$\psi_1(\alpha - x) = \int_0^{\alpha - x} \psi_1'(t)\, dt = \int_0^{\alpha} \psi_1'(t)\, dt + \int_{\alpha - x}^{\alpha} -\psi_1'(t)\, dt$$

$$> \int_0^{\alpha} \psi_1'(t)\, dt + \int_{\alpha}^{\alpha + x} \psi_1'(t)\, dt = \psi_1(\alpha + x)$$

Let us, now, consider some $x$. There are three possibilities: 1. $x \leq 2\alpha - 1$; 2. $x \in (2\alpha - 1, \alpha)$; and 3. $x \geq \alpha$. If $x \leq 2\alpha - 1$, then, since $y(x) \in (0, 1)$, it follows, trivially, that $x + y(x) < 2\alpha$.

If $x \in (2\alpha - 1, \alpha)$, let us set $\delta = \alpha - x$. Then, since $\psi_1$ is strictly convex, we have that $\psi_1(x) = \psi_1(\alpha - \delta) > \psi_1(\alpha)$. Consequently, by the Intermediate Value Theorem, there must exist some $y^* \in (\alpha, \alpha + \delta)$ such that $\psi_1(\alpha - \delta) = \psi_1(y^*)$. Clearly $\alpha - \delta \neq y^*$, and, therefore, since $\psi_1(\alpha - \delta) = \psi_1(y^*)$, it must be the case that $y(x) = y^*$. Furthermore, $x + y(x) = \alpha - \delta + y^* < \alpha - \delta + \alpha + \delta = 2\alpha$, and, therefore, our claim holds in that case as well.

Finally, if $x \geq \alpha$, let us set $\delta = x - \alpha$. Since $\alpha > 0.5$, we have that $\delta \in (0, 1 - \alpha)$, and therefore, $\psi_1(\alpha - \delta) < \psi_1(\alpha + \delta) = \psi_1(x)$. Furthermore, by the monotonicity of $\psi_1$ in the intervals $(\alpha - \delta, \alpha)$, $(\alpha, x)$ and $(x, 1)$, we also have, respectively, that

$$\psi_1(y) < \psi_1(\alpha - \delta) < \psi_1(x) \text{ for all } y \in (0, \alpha) \tag{11}$$

$$\psi_1(y) < \psi_1(x) \qquad\qquad \text{for all } y \in (\alpha, x) \tag{12}$$

$$\psi_1(y) > \psi_1(x) \qquad\qquad \text{for all } y \in (x, 1). \tag{13}$$

In other words, it has to be the case that $y(x) \in (0, \alpha - \delta)$, and, therefore, $x + y(x) < \alpha + \delta + \alpha - \delta = 2\alpha$.

It follows that, if $\alpha > 0.5$, $y(x) < 2\alpha - x$. We remark that the case of $\alpha < \frac{1}{2}$ follows identical arguments, while the case $\alpha = \frac{1}{2}$ is trivial. $\qquad\square$

**Remark C.2** (Technical intuition of Theorem 4.6). *It is important to provide the mathematical (technical) interpretation of Theorem 4.6. From equation QRD, we observe that the speed of the projection dynamics, ($q = 0$), does not depend on the current state, since the $x$ term is eliminated from the right hand side. As a result, the separatrix is a straight line, see e.g., panel 1 in Figure 1. As q increases for values between $0$ and $1$ for which our analysis applies, the dynamics leave the boundary must faster, cf. panels 2 to 4 in Figure 1, while being fixed between the three points, i.e., the corners $(0,1)$, $(1,0)$ and the mixed NE. This implies a certain curvature for the separatrix. On the other hand, the center of the simplex, i.e., the $(0.5, 0.5)$ point, belongs to the region of attraction of the risk-dominant equilibrium under all dynamics. Combining these two, we get that the replicator dynamics create a convex separatrix when risk- and payoff-dominant equilibria differ, and a concave separatrix when risk- and payoff-dominant equilibria coincide, cf. Appendix D. This provides the picture that we have, with the straight-line separatrix of the gradient descent (projection) dynamics being below the curved separatrix of the replicator when risk- and payoff-dominant equilibria differ and above when risk- and payoff-dominant equilibria coincide. The above explanation drives our current proof of Theorem 4.6. Deriving a game-theoretic interpretation (along with the technical one above) is an interesting open question.*

Finally, a direct (technical) implication from the proof of Theorem 4.6 is provided in Lemma C.3 which may be of independent interest. Recall from the proof of Theorem 4.6 that the solutions to $\Psi_1(x, y) = 0$ are the functions $y = x$ and $y : (0, 1) \to (0, 1)$ such that $y(\alpha) = \alpha$, and for all $x \in (0, 1) \setminus \{\alpha\}$ it holds that $y(x) \neq \alpha$, and $\psi_1(y(x)) = \psi_1(x)$.

**Lemma C.3** (Curvature of the stable manifold of RD). *Consider the $1$-replicator dynamics (RD) in the parametric game $\Gamma_{w,\beta}$. The stable manifold, $\mathcal{M}_{\text{Stable}}$ of RD in $\Gamma_{w,\beta}$ is given by the curve $y = y(x)$. If the payoff-dominant equilibrium, $x = y = 0$, is also risk-dominant, then $y$ is strictly concave. Conversely, if the non-payoff dominant equilibrium, $x = y = 1$, is risk-dominant, then $y$ is strictly convex. Otherwise, $y(x) = 1 - x$.*

*Proof of Lemma C.3.* By differentiating both sides of the implicit function $\psi_1(y(x)) = \psi_1(x)$ with respect to $x$, we get that $\psi_1'(y(x))y'(x) = \psi_1'(x)$, i.e., $y'(x) = \frac{\psi_1'(x)}{\psi_1'(y(x))}$. Notice that, since $y$ is bijective, $\psi_1'$ is monotonic ($\psi_1'' < 0$), and $\psi_1'(y(\alpha)) = \psi_1'(\alpha) = 0$, the above equality is well-defined for all $x \in (0, 1) \setminus \{\alpha\}$. Hence, we have that:

$$y'' = \frac{\psi_1''(x)[\psi_1'(y)]^2 - \psi_1'(x)\psi_1''(y)y'}{[\psi_1'(y)]^2} = \frac{\psi_1''(x)[\psi_1'(y)]^2 - [\psi_1'(x)]^2\psi_1''(y)}{[\psi_1'(y)]^3}$$
$$= \frac{y^3(1-y)^3\alpha(1-\alpha)(x-y)(y+x-2\alpha)}{(\alpha-y)^3x^2(1-x)^2y^2(1-y)^2},$$

where the dependency of $y$ to $x$ is implied for compactness. Hence, we have that $y''(x) < 0$ if and only if $\frac{(x-y)(y+x-2\alpha)}{\alpha-y} < 0$. However, by the Intermediate Value Theorem (IVT) applied on $\psi_1$ in $(0, \alpha)$, and $(\alpha, 1)$, and the definition of $y$, it follows, trivially, that $x \leq y$ if, and only if, $\alpha \leq y$, with equality in both inequalities only if $x = \alpha$. Therefore, $\frac{x-y}{\alpha-y} > 0$, $\forall x \in (0, 1) \setminus \{\alpha\}$; hence $y''(x) < 0$ if, and only if, $y + x - 2\alpha < 0$, which by the proof of Theorem 4.6 is equivalent to $\alpha > \frac{1}{2}$. This concludes the proof for the first statement. The second statement follows in a similar manner by requesting $y''(x) > 0$, while the last statement is trivial. $\qquad\square$

*Proof of Theorem 4.8.* Let $\Gamma_{w,\beta}$ be a $2 \times 2$ symmetric PRPG, where the payoff-dominant equilibrium, $x = y = 0$, is also risk-dominant, i.e., $\beta > 1 - w$, or equivalently $\alpha > 0.5$, where $x^* = y^* = \alpha$ is the mixed NE of $\Gamma_{w,\beta}$. Recall that, by equation 10, the stable manifold of the mixed NE of $\Gamma_{w,\beta}$ with respect to GD is the line segment $\ell : y = 2\alpha - x$, for $x \in (\max\{0, 2\alpha - 1\}, \min\{2\alpha, 1\})$. Since $\alpha > 0.5$, we have that $2\alpha - 1 \geq 0$ and $2\alpha \geq 1$; therefore, the extreme points of $\ell$ are $(0, 2\alpha)$ and $(2\alpha, 0)$. That implies the the RoA of $(1, 1)$ is the triangle with extreme points at $(1, 1)$, $(2\alpha - 1, 1)$, and $(1, 2\alpha - 1)$. Since that is a right triangle, with both its base and its height equal

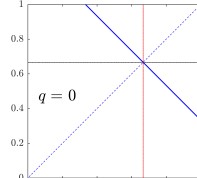 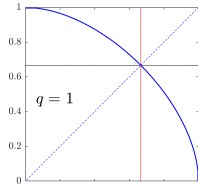 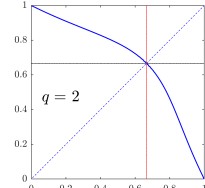 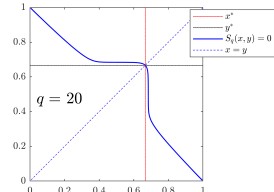

**Figure 5:** The stable manifolds, $\Psi_{q,\text{Stable}}(x, y) = 0$, (solid blue lines) for the same values of $q$ and the same instance of $\Gamma_{w,\beta}$ as in Figure 1, in which the payoff- and risk- dominant NE is at the bottom left corner. For all $q$, the separatrix goes through the mixed NE at the intersection of the $x^*$ (dashed red) and $y^*$ (dashed black) coordinates. All panels also include the unstable manifold defined by $x - y = 0$ (dashed blue line). The region of attraction of the payoff-dominant NE is larger for all values of $q$; however, this is because this NE is also risk-dominant, cf. Theorem 4.6.

to $2(1 - \alpha)$, the Lebesque measure of RoA$(0,0)$ is $\mu(\text{RoA}(1,1)) = 2(1 - \alpha)^2$; subsequently, $\mu(\text{RoA}(0,)) = 1 - \mu(\text{RoA}(1,1)) = 1 - 2(1 - \alpha)^2$. We may calculate the APoA of GD in $\Gamma_{w,\beta}$ as a function of $w$ and $\beta$. Specifically, when $\alpha > 0.5$, i.e., $\beta < 1 - w$, we have that:

$$
\begin{aligned}
\text{APoA}(w, \beta) &:= \text{APoA}(\text{GD}, \Gamma_{w,\beta}) = \frac{\max_{x,y \in [0,1]} \text{SW}(x, y)}{\text{APM}_{\text{SW},[0,1]^2}(\text{GD}, \Gamma_{w,\beta})} \\
&= \frac{\text{SW}(0, 0)}{\text{SW}(0, 0) \cdot \mu(\text{RoA}(0,0)) + \text{SW}(1, 1) \cdot \mu(\text{RoA}(1,1))} \\
&= \frac{w}{w\mu(\text{RoA}(0,0)) + \mu(\text{RoA}(1,1))} \\
&= \frac{w}{w[1 - 2(1 - \alpha)^2] + 2(1 - \alpha)^2} = \frac{w(w + 1 - \beta)^2}{w(w + 1 - \beta)^2 - 2(w - 1)(1 - \beta)^2}.
\end{aligned}
$$

where we used that $\alpha = 1 - \left(\frac{1 - \beta}{w + 1 - \beta}\right)^2$. We may, now, perform a first-order analysis in APoA$(w, \beta)$; that is, for all $\beta \geq 1 - w$, we have that:

$$
\frac{\partial \text{APoA}(w, \beta)}{\partial \beta} = \frac{-4w^2(w - 1)(w + 1 - \beta)(1 - \beta)}{[w(w + 1 - \beta)^2 - 2(w - 1)(1 - \beta)^2]^2} \leq 0.
$$

From the above, it follows that APoA$(w, \beta) \leq$ APoA$(w, 1 - w)$; that is:

$$
\text{APoA}(w, \beta) \leq \frac{4w^3}{4w^3 - 2(w - 1)w^2} = \frac{2w^3}{w^3 + w^2} < 2,
$$

where the last inequality follows by letting $w \to \infty$. Notice that this bound is tight. It is not difficult to see that if $\alpha < 0.5$, APoA$(w, \beta)$ is unbounded. □

## D  VISUALIZATIONS: INVARIANT FUNCTIONS AND SEPARATING MANIFOLDS

In this part, we provide systematic, and essentially exhaustive, visualizations of the stable manifolds (separatrices) in the $\Gamma_{w,\beta}$ class.

**Same payoff- and risk-dominant NE in $\Gamma_{w,\beta}$.**  As mentioned in the main part, the separatrix for different values of $q$ can be obtained by plotting the 0-level set of the invariant functions in Figure 1. These are depicted in Figure 5. As a sanity check, we also see from Figure 5 that the region of attraction of the payoff-dominant equilibrium for $q = 0$ (GD dynamics) is larger than the region of attraction for $q = 1$ (RD).

**Stable manifolds for all $q \geq 0$.**  In a similar vein to Figure 2, we next depict the separatrices, stacked for all values of $q \in [0, 10]$, for different parameterizations of the $\Gamma_{w,\beta}$ class (Figure 6. In all panels of Figure 6, parameter $w$ is equal to 2. We obtain qualitatively equivalent plots for any $w > 1$ and $\beta$ small enough. The main takeaways from the (essentially exhaustive) visualizations in the panels of Figure 6 are that (i) the region of attraction of the risk-dominant equilibrium is

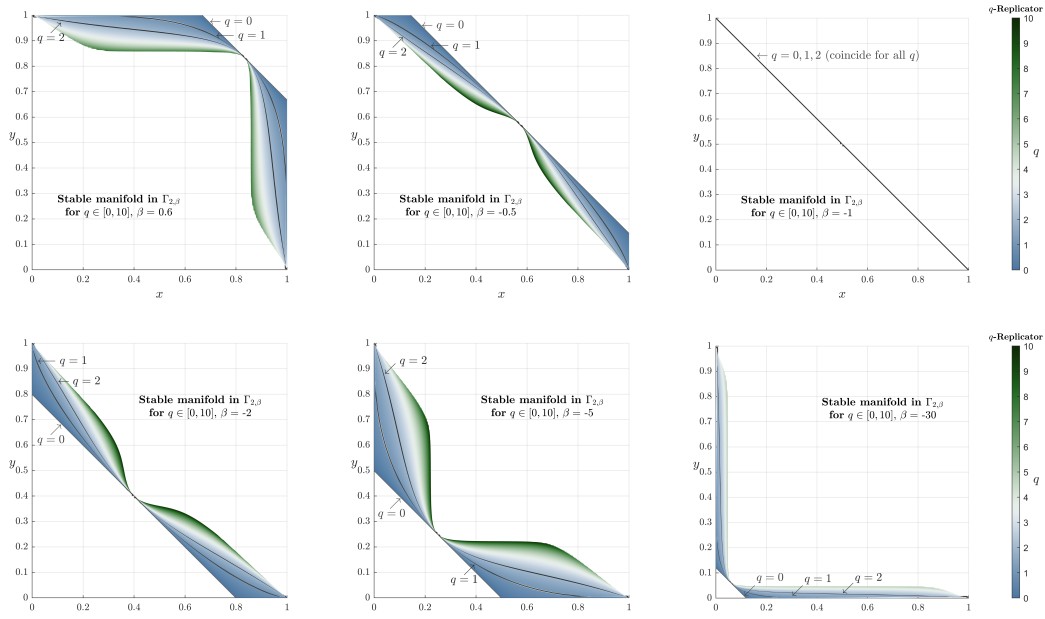

**Figure 6:** Stable manifolds (separatrices) for all different values of $q \in [0, 10]$ (from blue to brown) in different parameterizations of the $\Gamma_{w,\beta}$ game for $w = 2$ and varying $\beta$. In all panels, the manifolds for $q = 0$, $q = 1$, and $q = 2$ are shown in shades of black for reference. The region of attraction of the payoff-dominant equilibrium (bottom-left corner) shrinks as $q$ increases when this equilibrium is also risk-dominant ($\alpha > 0.5$) and increases with $q$ when this equilibrium is *not* risk-dominant ($\alpha < 0.5$). In fact, as $\beta$ decreases, the payoff-dominant becomes increasingly more "risky" and its region of attraction becomes arbitrarily small.

larger for all $q \geq 0$ regardless of whether this equilibrium is payoff-dominant or not, (ii) the region of attraction of the payoff-dominant equilibrium may become arbitrarily small as this equilibrium becomes arbitrarily risky. In particular, observation (ii), suggests that in this case, it is hopeless to bound any static or average performance measure. This became more transparent with the APoA analysis in the previous Section of the Appendix (cf. Theorem 4.8 in the main paper). We conclude this part with some visualizations of the stable and unstable manifolds in a $2 \times 2$ non-symmetric PRPG.

**non-symmetric PRPGs.** Consider the identical-interest PRPG, $\mathrm{ID}_{w,\beta}$, with identical payoff functions $u_{w,\beta,1}(s_1, s_2) = u_{w,\beta,2}(s_1, s_2) = A_{w,\beta,s_1,s_2}$, where the payoff matrix $A_{w,\beta} \in \mathbb{R}^{2\times 2}$ is given by $A_{w,\beta} = \begin{pmatrix} 1 & 0 \\ \beta & w \end{pmatrix}$, $\beta \leq 1 \leq w$. The game $\mathrm{ID}_{w,\beta}$ has the same pure NE as the games $\Gamma_{w,\beta}$, namely $x = y = 0$, with payoff $w$, and $x = y = 1$, with payoffs 1 for both players, but now the mixed NE is not symmetric and it is given by $x^* = (w - \beta)/(w + 1 - \beta)$ and $y^* = w/(w + 1 - \beta)$.

In Figure 7, we visualize the stable *and* unstable manifolds for all values of $q \in [0, 10]$ in an instance of $\mathrm{ID}_{w,\beta}$ with $w = 2$ and $\beta = -2$. In this case, the separating (stable) manifolds do not increase (decrease) monotonically with $q$ as it is evident from the overlapping (equally) colored regions. Thus, it requires a different approach to estimate whether the size of the regions of attraction of the payoff-dominant equilibrium follow a certain monotonicity pattern, which again may change depending on whether this equilibrium is also risk-dominant or not. In the context of the current paper, Figure 7 highlights that (i) the geometry of the regions of attraction is highly complex under different algorithms (parametrizations of QRD) even for low-dimensional, identical interest games, and (ii) given this complexity, the findings in the class $\Gamma_{w,\beta}$ become even more surprising and intriguing.

**Different payoff- and risk-dominant NE in $\Gamma_{w,\beta}$.** The main differences in the class $\Gamma_{w,\beta}$ occur between games in which the payoff- and risk-dominant equilibria coincide and games in which they differ. Recall that Figure 1 shows the invariant function in a $\Gamma_{w,\beta}$ instance, where $w = 2$ and $\beta = 0$,

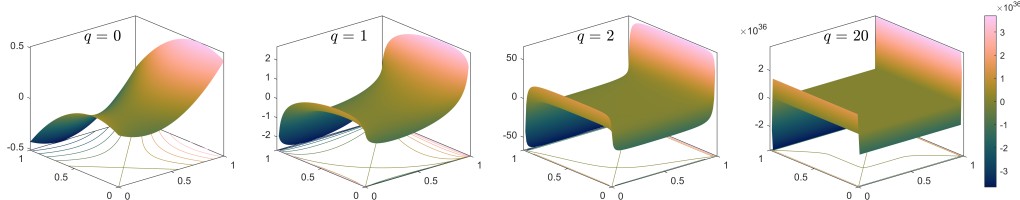

**Figure 8:** The invariant function, $\Psi_q(x, y)$, for all $x, y \in [0, 1]^2$ in the game $\Gamma_{w,\beta}$ for $w = 2$, $\beta = -4$, and the same values of $q$ as in Figure 1: $q = 0$ (gradient descent dynamics), $q = 1$ (standard replicator dynamics), $q = 2$ (log-barrier dynamics), and $q = 20$. The invariant function again becomes very steep at the boundary as $q$ increases, taking both arbitrarily large negative (**dark**) and positive (light) values in the vicinity of the NE.

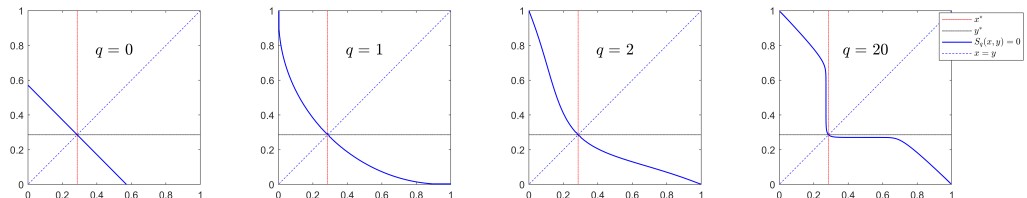

**Figure 9:** The stable manifolds, $\Psi_{q,\text{Stable}}(x, y) = 0$, (solid blue lines) for the same values of $q$ and instance of $\Gamma_{w,\beta}$ as in Figure 8, in which the payoff-dominant NE is at the bottom-left corner and the risk-dominant NE is at the upper-right corner. For all $q$, the separatrix goes through the mixed NE at the intersection of the $x^*$ (dashed red) and $y^*$ (dashed black) coordinates. All panels also include the unstable manifold defined by $x - y = 0$ (dashed blue line). The region of attraction of the payoff-dominant NE is now smaller for all $q$, because this NE is not risk-dominant, cf. Theorem 4.6.

i.e., in which the payoff- and risk- dominant equilibria coincide. In Figure 8, we provide an instance in which the payoff- and risk-dominant equilibria differ. Similar to Figure 5, Figure 9 depicts the separating manifolds (stable manifolds or separatrices) of the regions of attractions of the two pure NE. These are precisely the zero-level sets of the invariant functions shown in Figure 8. As we can see, in this case, the region of attraction of the payoff-dominant equilibrium, $w$ is smaller than the region of the, now, risk-dominant equilibrium, $1$. Intuitively, when a NE becomes risk-dominated, its region of attraction shrinks, even if this NE is payoff-dominant. This is because, for a mixed choice of distributions, the risk-dominant NE yields a higher utility and is that more "attractive" for the dynamics. This trade-off between high reward at a certain state (e.g., $w, w$) and high risk if that state is not reached (e.g., $\beta, 0$, with $\beta < 0$), also explains why socially optimal, but otherwise risky, outcomes, e.g., the adoption of revolutionizing technology or a social norm that challenges the status quo, are never reached in real-life situations.

**On risk-dominance in higher dimensions** In diagonal games (D-PRPGs), it is straightforward to make an equilibrium more *risky*. This is achieved simply by replacing the zero entries in the corresponding line of matrix $U$ by some negative number, e.g.,

$$U_{\text{risky},m} = \begin{pmatrix} u_1 & 0 & \ldots & 0 \\ 0 & u_2 & \ldots & 0 \\ \vdots & \vdots & \ddots & \vdots \\ -10 & -10 & \ldots & u_m \end{pmatrix} := \text{diag}(u_1, u_2, \ldots, u_m; \text{risk}_n = -10)$$

In this case, the payoff-dominant equilibrium with payoff $u_m$ to each player becomes more risky, since a failure to coordinate on it results to a negative payoff of $-10$ for the agent who selected the corresponding action. Proceeding in a similar fashion, one may replace the zero entries with an (arbitrarily large) negative element in all lines of the matrix except for the first one. In analogy to the $U_{\text{risky},m}$ notation, we will denote such games by $U_{\text{risky}} := \text{diag}(u_1, u_2, \ldots, u_m; \text{risk} = -r)$, where $r > 0$ is the *risk constant* (equal to $-10$ in the example above). In this way, all equilibria become more risky except for the payoff-dominated one, i.e., the equilibrium with payoffs $u_1$ to each agent which corresponds to the action profile in which every agent selects their first action.[2] Concerning

---

[2] An alternative interesting approach to generalize the notion of risk-dominance in arbitrary games is via pairwise comparisons of actions as proposed by Honda (2012).

our experiments, the outcome of Figure 4 is reversed in games of the form $U_{\text{risk}}$, i.e., in games in which the payoff superior equilibria were more risky (not reported here).

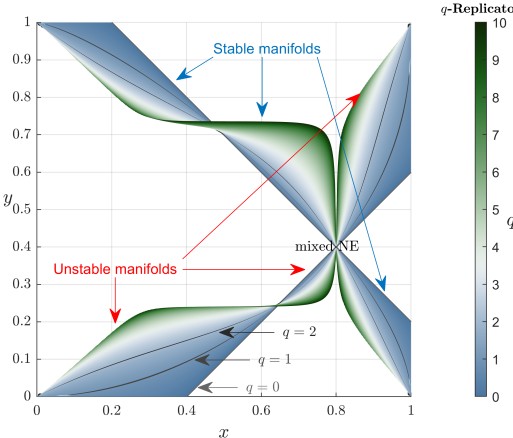

**Figure 7:** Stable and unstable manifolds for all $q \in [0, 10]$ in an instance of the identical interest game $\text{ID}_{w,\beta}$ with $w = 2$ and $\beta = -2$. The unique mixed NE is not symmetric and lies at $x^* = 0.8, y^* = 0.4$. The main difference with the symmetric games in $\Gamma_{w,\beta}$ is that the regions of attraction of the payoff-dominant equilibrium (bottom-left corner) are not increasing (nor decreasing) in $q$ anymore.

