# OpenReview forum: "Beating Price of Anarchy and Gradient Descent without Regret in Potential Games"
_ICLR.cc/2024/Conference — ICLR 2024 poster_

### Official Review · Reviewer_ycpS · 2023-10-29

**Soundness:** 4 excellent
**Presentation:** 3 good
**Contribution:** 3 good
**Rating:** 8
**Confidence:** 3

**Summary:**

This paper studies the convergence properties of q-replicator dynamics in continuous-time potential games. Potential games are a special class of general-sum games in which player utilities can be represented by a potential function. q-replicator dynamics are a class of learning dynamics which take a hyperparameter $q$ as input. If $q=0$ gradient descent is recovered, if $q=1$ replicator dynamics are recovered, and if $q=2$ log-barrier dynamics are recovered. The authors show that q-replicator dynamics converge pointwise to Nash equilibria in a subclass of potential games called Perfectly-Regular Potential Games (PRPGs), which capture almost all finite potential games.

Having established this result, the authors are interested in analyzing which equilibria are converged to when all players in a game employ q-replicator dynamics. Their main metric of performance is the average price of anarchy (APoA), which measures the price of anarchy, on average, with respect to a uniformly random initialization of the q-replicator dynamics. The authors' main theoretical contributions come in the setting of 2-by-2 PRPGs. In particular, they find that the APoA of running gradient descent is higher than the APoA of running replicator dynamics in 2-by-2 PRPGs if and only if the payoff-dominant equilibrium is also risk-dominant. Here, the "payoff-dominant" equilibrium refers to the equilibrium with the highest utility, and an equilibrium is "risk-dominant" if it is unilaterally optimal against the uniform distribution of the rest of the agents.

The authors compliment their theoretical results in 2-by-2 PRPGs with numerical simulations in higher dimensions. They find that, in line with their theoretical results in 2-by-2 games, gradient descent outperforms the replicator dynamics in all diagonal games (i.e. games in which the only non-zero elements of the payoff matrix are along the diagonal). Throughout the submission, the authors provide proof sketches and intuition for their results.

**Strengths:**

The problem of analyzing the convergence guarantees of learning algorithms in games is not a novel problem, but is of great interest to the game theory community. Additionally, obtaining results in this area has traditionally been challenging. While the authors restrict themselves to a subclass of two-by-two potential games, the authors' results are novel and their analysis helps shed some light on this important issue by comparing the convergence properties of gradient descent and replicator dynamics, two canonical algorithms for learning in games. I found the paper to be well-written, and I appreciated the authors' efforts to provide intuition for their results and why they cannot be easily extended beyond the two-by-two setting they consider. The numerical simulations in settings beyond two-by-two games are also a strength, as they help shed light on this problem in a more general setting, despite the issues with extending the authors' theoretical results.

**Weaknesses:**

One weakness is the restriction to the setting of 2-by-2 Perfectly-Regular Potential Games. However given the challenge of analyzing the convergence dynamics of learning algorithms in games, this is not a large weakness.

**Questions:**

Why focus on metrics which measure performance with respect to random initialization instead of other methods of initialization, e.g. initializing to the uniform distribution over actions as is canonical in the literature on learning in games?

Do you have any insights on how your results would carry over to the discrete-time setting?

---

> ### Author Response · Authors · 2023-11-17
> **Response to Reviewer ycpS**
>
> We are delighted to hear that you found our paper to be well-written, and that you appreciated our efforts to provide intuition for our results!
>
> - **Question 1:** *Why focus on metrics which measure performance with respect to random initialization instead of other methods of initialization, e.g. initializing to the uniform distribution over actions as is canonical in the literature on learning in games?*
>
> - **Response**:
>     - Firstly, as we show, starting from the uniform distribution, the dynamics are guaranteed to converge to the risk-dominant equilibrium which, unless it coincides with the payoff dominant equilibrium, can be detrimental to social welfare. Thus, the convention to start at the uniform distribution is, in fact, a restriction rather than a desirable norm.
>     - Secondly, we believe that our quantification of uncertainty is more appropriate for our purpose. Specifically, if we assume that all the agents always start from a uniform distribution that seems to convey a belief from the perspective of the experimenter that the related individuals do not have any inherent biases for choosing one action versus another. But in practice there could be a lot of conscious or unconscious biases at play; for instance, due to how the order with which the actions are presented to the participants, or the names given to the actions, etc. Thus, we believe that it is more appropriate to encode a type of epistemic uncertainty that means that we are not being certain about what the relevant probability distribution is (see also the Wikipedia discussion [here]( https://en.wikipedia.org/wiki/Uncertainty_quantification)). *Critically, this was also the choice in the original paper by Panageas and Piliouras that we are building upon.*
>     - Finally, perhaps it is also useful to point out why our results are robust to the choice of the initial measure which does not have to be exactly the uniform measure, i.e., the Lebesgue measure. Specifically, we could easily allow for the measure of each set to be within some fixed multiplicative constant, $\gamma$, of its true Lebesgue measure, in which case, the probability of converging to the good/bad equilibrium could only be amplified/reduced by at most $\gamma$. Combining this with the analysis for the uniform case, one could then derive such APoA bounds, but now they would be parametric on $\gamma$.
> - **Question 2:** *Do you have any insights on how your results would carry over to the discrete-time setting?*
> - **Response:** The details of the analysis would depend on the exact choice of discretization, e.g., whether you discretize replicator as Multiplicative Weight Updates (i.e. the Euler method in the dual space), or via a discretization in the primal space, or e.g., via discrete replicator dynamics also known as linear MWU). Generally there could be shuttle differences in the behavior of a given orbit given the exact choice of discretization (see [1]). Such subtleties aside, a good rule of thumb is that by choosing small enough step-size the regions of attraction of the discretized systems can effectively become indistinguishable from the continuous-time one. Pushing such approaches towards understanding larger steps sizes, which could destabilize some discretizations into chaos (again see [1]) is an interesting direction but beyond the scope of this work.
>     1. Palaiopanos, Gerasimos, Ioannis Panageas, and Georgios Piliouras. "Multiplicative weights update with constant step-size in congestion games: Convergence, limit cycles and chaos." Advances in Neural Information Processing Systems 30 (2017).
>
> Thank you for the interesting questions!

---

### Official Review · Reviewer_xUnV · 2023-11-01

**Soundness:** 4 excellent
**Presentation:** 3 good
**Contribution:** 3 good
**Rating:** 6
**Confidence:** 2

**Summary:**

A long line of work in algorithmic game theory has sought to bound the difference between selfish outcomes and socially optimal outcomes. However, these ``Price of Anarchy" (PoA) bounds hold in a worst-case sense, and need not faithfully represent the real-world performance of concrete dynamics. This work considers the average-case PoA (APoA) of a large family of dynamics, namely Q-Replicator dynamics, in potential games. The main contributions of this work are threefold: first, it is shown that almost all potential games are such that Q-replicator dynamics initialized on any interior point converges to a (finite set of) Nash equilibrium. In particular, this makes average-case notions of PoA almost always well-defined. In the case of 2x2 games, this paper provides a complete characterization of the regions of attraction for different equilibria. The main applications are to exactly characterize in which 2x2 games gradient descent dynamics has a better APoA than replicator dynamics, and moreover, bound the APoA of gradient descent when the payoff- and risk-dominant Nash equilibria coincide. Finally, preliminary empirical evidence is given to suggest that certain generalizations of the results are possible.

**Strengths:**

The conceptual questions addressed by this work are quite interesting. On a technical level, the analysis provides a nontrivial extension of existing work on the performance of these dynamics. The paper is itself quite well-written and the results are supplemented with empirical evaluations that validate the results and suggest intriguing future directions.

**Weaknesses:**

The theoretical results, while very nontrivial, currently only hold for 2x2 games, and it is not clear conceptually or technically what an appropriate generalization would be. The paper could benefit from a bit more discussion about the technical differences with prior work (i.e. that of Panageas and Piliouras [EC'16]).

**Questions:**

Recommendation: While I only have a passing familiarity with this line of work, my general impression is that work has very interesting conceptual takeaways. While there is some concern about how general these findings are and the extent to which they might generalize, my current belief is that this work makes interesting progress on the quality of important learning outcomes that would be of interest to the community. For now, I am leaving my score as a 6, but I could increase it in light of other reviewer comments/discussion of the technical novelty compared to Panageas and Piliouras [EC'16].

Comments:

---The conceptual goal of providing more refined PoA guarantees for concrete dynamics is a very interesting direction that deserves further study, in my opinion.

---While this work does give good discussion on how the current paper extends the original results of Panageas and Piliouras, it may be worth providing a short discussion on the technical differences that allow these extensions. This would also help convince the reader of the novelty of these results.

---As mentioned above, the main weakness is that the concrete theoretical results on APoA are localized to 2x2 games. It isn't entirely clear what an appropriate generalization would look like with more agents and larger action spaces, and of course, one imagines that the technical analysis quickly becomes intractable. But progress on these questions has to start somewhere and my current impression is that this paper indeed takes steps towards these aims that would be of value to the community.

---

> ### Author Response · Authors · 2023-11-17
> **Response to Reviewer xUnV**
>
> We are very happy to hear that you found the conceptual questions addressed by our work interesting and suggestive of intriguing future directions!
>
> *Suggestion: While this work does give good discussion on how the current paper extends the original results of Panageas and Piliouras, it may be worth providing a short discussion on the technical differences that allow these extensions. This would also help convince the reader of the novelty of these results... A bit more discussion about the technical differences with prior work (i.e. that of Panageas and Piliouras [PP]).*
>
> **Response:** We will be happy to expand a bit further on our technical differences between our paper and the precursor one by Panageas and Piliouras 2016 ([PP16]). What [PP16] show is that the APoA is effectively well-posed for all linear congestion games using a rather elaborate but hard to generalize argument. This argument critically relies on the exact definition of replicator dynamics as well as on the linearity of costs. Instead, our approach follows a different proof technique that allows us to generalize along both the game and the learning dynamics dimensions. We can break down our technical differences in three parts.
>
> - **Well-posedness of APoA:** Our first technical contribution is that our analysis allows for APoA to be effectively well-defined and applied to all generic potential games for all q-replicator dynamics (Section 3). The related techniques have no analogue in the original paper and leverage recent advances in the theory of potential games (Swenson 2020). This is the convergence result of Theorem 3.1. Such approaches that target generic (versus all games) within some class of games are commonly used in game dynamics (see e.g. Theorem 3.3 in [1]) or the quasi-strict genericity assumption in [2-5].
>
>     1. Bailey, James P., and Georgios Piliouras. "Multiplicative weights update in zero-sum games." Proceedings of the 2018 ACM Conference on Economics and Computation. 2018.
>
>     2. Vlatakis-Gkaragkounis, Emmanouil-Vasileios, et al. "No-regret learning and mixed nash equilibria: They do not mix." Advances in Neural Information Processing Systems 33 (2020): 1380-1391.
>
>     3. Johanne Cohen et al  Learning with bandit feedback in potential games. In NIPS’17: Proceedings of the 31st International Conference on Neural Information Processing Systems, 2017.
>
>     4. Laraki, Rida, et al. Mathematical foundations of game theory. New York, NY, USA: Springer, 2019.
>
>     5. Fudenberg, Drew, and Jean Tirole. Game theory. MIT press, 1991.
>
> - **Average APoA analysis:** Regarding the dynamics dimension, our second technical contribution is that we generalize the analysis from replicator to a broad class of related dynamics. Our approach extends the invariant function technique, see Lemma 4.7, which enables the proof of Theorem 4.6. Despite the thematic similarity between this part and the related part in the [PP] paper, there is also a lot of new technical content on the exact nature of the separatrix and how it affects the dynamics. For instance, one such result is captured in Lemma B.3 (supplementary) which concerns the curvature of the separatrix. Such results inform our intuition on the reasons why our main results hold (for instance Theorem 4.6) and we believe that they can enable a lot of interesting follow-up work. We also believe that our proof technique is flexible enough to be applied on different classes of learning dynamics. Other similar papers that generalize findings from replicator like dynamics to other related dynamics are: from [1] to [2] and from [3] to [4].
>
>     1. Piliouras, Georgios, and Jeff S. Shamma. "Optimization despite chaos: Convex relaxations to complex limit sets via Poincaré recurrence." Proceedings of the twenty-fifth annual ACM-SIAM symposium on Discrete algorithms. Society for Industrial and Applied Mathematics, 2014
>
>     2. Mertikopoulos, Panayotis, et al. "Cycles in adversarial regularized learning." Proceedings of the twenty-ninth annual ACM-SIAM symposium on discrete algorithms. Society for Industrial and Applied Mathematics, 2018.
>
>     3. Palaiopanos, Gerasimos, et al. "Multiplicative weights update with constant step-size in congestion games: Convergence, limit cycles and chaos." Advances in Neural Information Processing Systems 30 (2017).
>
>     4. Bielawski, Jakub, et al. "Follow-the-regularized-leader routes to chaos in routing games." International Conference on Machine Learning. PMLR, 2021.
>
>  - **Experiments:** Finally, our paper has an extensive experimental section (along with accompanying code) that as the Reviewer acknowledges raises some very intriguing directions for future work. No such section or software package was made available in [PP].
>
> We hope that the above clarify the technical differences between the two papers. Thank you again for your support and helpful suggestions!

---

### Author Response · Authors · 2023-11-17

We are grateful to our reviewers for the time they invested in reading our paper as well as for their helpful comments and suggestions.
We are very encouraged by their warm support!

---

### Meta-Review · Area_Chair_dwZ3 · 2023-12-17

**Metareview:**

This paper considers the performance of iterative-update algorithms for user updates in standard game-theoretic setting, and proves strong and novel results on the convergence to and (sub)optimality of the resulting fixed points.


Strength: clear writing, non-trivial new theoretical results. it is the first paper to directly connect the choice of the iterative algorithm with the quality of the final fixed point in these game settings.
Weakness: possibly a niche topic for ICLR

Even though only two reviewers managed to return the reviews, these were relatively detailed and there was quality discussion.

**Justification For Why Not Higher Score:**

Possibly a niche topic for ICLR

**Justification For Why Not Lower Score:**

Significant novelty and high reviewer scores.

---

### Decision · Program_Chairs · 2024-01-16

Accept (poster)